# Automated Knowledge Concept Annotation and Question Representation Learning for Knowledge Tracing

## Abstract

Knowledge tracing (KT) is a popular approach for modeling students' learning progress over time, which can enable more personalized and adaptive learning. However, existing KT approaches face two major limitations: (1) they rely heavily on expert-defined knowledge concepts (KCs) in questions, which is time-consuming and prone to errors; and (2) KT methods tend to overlook the semantics of both questions and the given KCs. In this work, we address these challenges and present KCQRL, a framework for *automated knowledge concept annotation and question representation learning* that can improve the effectiveness of any existing KT model. First, we propose an automated KC annotation process using large language models (LLMs), which generates question solutions and then annotates KCs in each solution step of the questions. Second, we introduce a contrastive learning approach to generate semantically rich embeddings for questions and solution steps, aligning them with their associated KCs via a tailored false negative elimination approach. These embeddings can be readily integrated into existing KT models, replacing their randomly initialized embeddings. We demonstrate the effectiveness of KCQRL across 15 KT models on two large real-world Math learning datasets, where we achieve consistent performance improvements.

## 1 Introduction

The recent years have witnessed a surge in online learning platforms (Adedoyin & Soykan, 2023; Gros & García-Peñalvo, 2023), where students learn new knowledge concepts, which are then tested through exercises. Needless to say, personalization is crucial for effective learning: it allows that new knowledge concepts are carefully tailored to the current knowledge state of the student, which is more effective than one-size-fits-all approaches to learning (Cui & Sachan, 2023; Xu et al., 2024). However, such personalization requires that the knowledge of students is continuously assessed, which highlights the need for ***knowledge tracing (KT)***.

In KT, one models the temporal dynamics of students' learning processes (Corbett & Anderson, 1994) in terms of a core set of skills, which are called *knowledge concepts (KCs)*. KT models are typically time-series models that receive the past interactions of the learner as input (e.g., her previous exercises) in order to predict response of the learner to the next exercise. Early KT models were primarily based on logistic or probabilistic models (Corbett & Anderson, 1994; Cen et al., 2006; Pavlik et al., 2009; Käser et al., 2017; Vie & Kashima, 2019), while more recent KT models build upon deep learning (Piech et al., 2015; Abdelrahman & Wang, 2019; Long et al., 2021; Liu et al., 2023b; Huang et al., 2023; Zhou et al., 2024; Cui et al., 2024).

Yet, existing KT models have two main limitations that hinder their applicability in practice (see Fig. 1). ① They require a comprehensive mapping between KCs and questions, which is typically done through manual annotations by experts. However, such KC annotation is both time-intensive and prone to errors (Clark, 2014; Bier et al., 2019). While recent work shows that LLM-generated KCs may be more favorable for human subjects (Moore et al., 2024), its success has not yet translated into improvements in KT. ② KT models overlook the semantics of both questions and KCs. Instead, they merely treat them as numerical identifiers, whose embeddings are randomly initialized and are learned throughout training. Therefore, existing KT models are expected to "implicitly" learn the association between questions and KCs and their sequential modeling for student histories, simultaneously. In

Figure 1: Overview of standard KT formulation and its limitations.

this paper, we hypothesize – and show empirically – that both ① and ② are key limitations that limit the predictive performance.

In this work, we propose a novel framework for *automated **k**nowledge **c**oncept annotation and **q**uestion **r**epresentation **l**earning*, which we call **KCQRL**[1]. Our KCQRL framework is flexible and can be applied on top of any existing KT model, and we later show that our KCQRL consistently improves the performance of state-of-the-art KT models by a clear margin. Importantly, our framework is carefully designed to address the two limitations ① and ② from above. Technically, we achieve this through the following three modules:

1. We develop a novel, automated KC annotation approach using large language models (LLMs) that both generates solutions to the questions and labels KCs for each solution step. Thereby, we effectively circumvent the need for manual annotation from domain experts (→ limitation 1).

2. We propose a novel contrastive learning paradigm to jointly learn representations of question content, solution steps, and KCs. As a result, our KCQRL effectively leverages the semantics of question content and KCs, as a clear improvement over existing KT models (→ limitation 2).

3. We integrate the learned representations into KT models to improve their performance. Our KCQRL is flexible and can be combined with any state-of-the-art KT model for improved results.

Finally, we demonstrate the effectiveness of our KCQRL framework on two large real-world datasets curated from online math learning platforms. We compare 15 state-of-the-art KT models, which we combine with our KCQRL framework. Here, we find consistent evidence that our framework improves performance by a large margin.

## 2    PRELIMINARIES: STANDARD FORMULATION OF KNOWLEDGE TRACING

**Knowledge tracing:** We consider the standard formulation of KT (e.g., Piech et al., 2015; Sonkar et al., 2020; Zhou et al., 2024), which is the performance prediction of the next exercise for a student based on time-series data (see Fig. 1). Specifically, for each student, the history of $t$ exercises are modeled as $\{e_i\}_{i=1}^t$ with exercises $e_i$. Each exercise $e_i$ is a 3-tuple, i.e. $e_i = \{q_i, \{c_{i,j}\}_{j=1}^{N_{q_i}}, r_i\}$, where $q_i \in \mathbb{Q}$ is the question ID, $c_{i,j} \in \mathbb{C}$ is a KC ID of $q_i$, and $r_i \in \{0, 1\}$ is the binary response (=incorrect/correct) of student $s$ for exercise $e_i$.

The aim of a KT model $F_\theta$ is to predict the binary response $\hat{r}_{t+1}$ of a student given the history of exercises and information about the next exercise. We denote the predicted response by $\hat{r}_{t+1} = F_\theta\big(q_{t+1}, \{c_{t+1,j}\}_{j=1}^{N_{q_{t+1}}}, \{e_i\}_{i=1}^t\big)$.

**Limitations:** In practice, the above KT formulation has two key limitations: ① It requires thousands of questions from $\mathbb{Q}$ to be manually annotated with each relevant KC from $\mathbb{C}$ among hundreds of categories Choi et al. (2020b); Wang et al. (2020); Liu et al. (2023c). Domain experts must ensure the consistency of manual annotations, a process that is not only resource-intensive but also prone to human error (Moore et al., 2024). ② Existing KT models overlook the semantic context of both questions and knowledge concepts (KCs), which weakens their ability to model learning sequences effectively. Typically, KT models initialize random embeddings $\mathrm{Emb}(q_i)$ for each question $q_i \in \mathbb{Q}$ and/or $\mathrm{Emb}(c_{i,j})$ for each KC $c_{i,j} \in \mathbb{C}$ (Abdelrahman et al., 2023; Zhu et al., 2023). The embeddings are then trained as part of the supervised learning objective of the KT task. As a result, the parameters $\theta$ of a KT model $F_\theta$ are trained simultaneously for both (a) learning the relationships between questions and KCs and (b) sequential dynamics of student learning histories.

---

[1]Our code is available at https://anonymous.4open.science/r/KCQRL and also in the supplementary material. Upon acceptance, we will move our codes to a public GitHub repository.

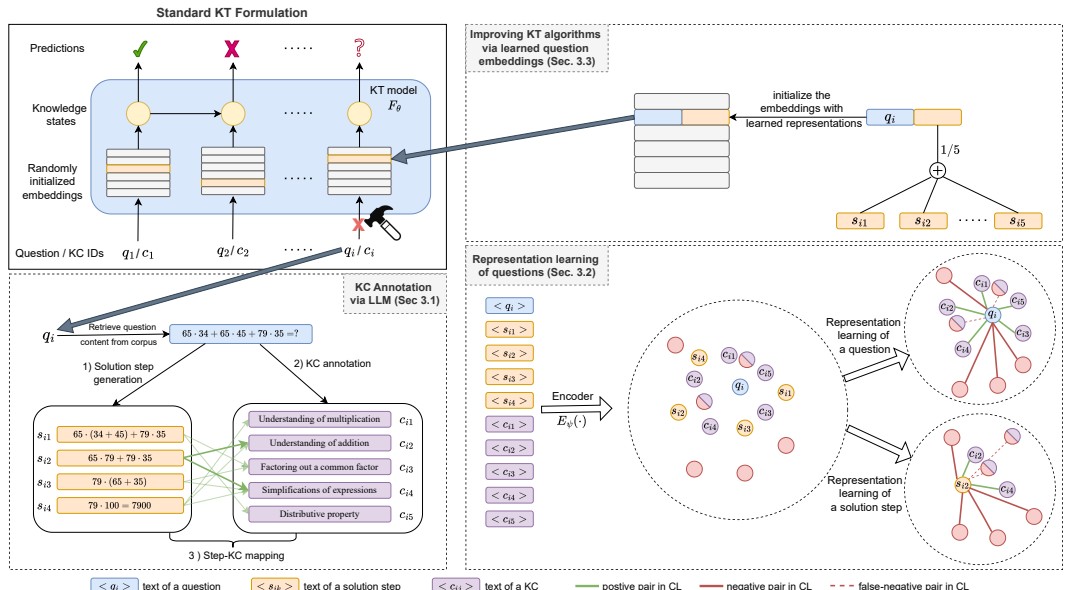

Figure 2: **Overview of our KCQRL framework and how it can be applied on top of existing KT models.** *Top left:* simplified illustration of the standard KT formulation, where the embeddings of questions and/or KC identifiers are initialized randomly for the prediction task. Our KCQRL improves the standard KT formulation via three modules: **(1)** *Bottom left:* shows how question IDs are translated into question content, solution steps (simplified for readability), and KCs via KC annotation (Sec. 3.1). **(2)** *Bottom right:* shows how these annotations are leveraged for representation learning of questions via a tailored contrastive learning and false negative elimination (Sec. 3.2). **(3)** *Top right:* shows how these learned representations initialize the embeddings of a KT model to improve the performance of the latter (Sec. 3.3).

**Proposed KT formulation:** In our work, we address the limitations ①and ②from above, and for this purpose, we propose a new formulation of the KT task. Specifically, our framework proceeds as follows: 1) We propose an automated KC annotation framework (Sec. 3.1) using chain-of-thought prompting of large language models (LLMs). Thereby, we effectively circumvent the need for manual annotation by domain experts ($\rightarrow$ limitation 1). (2) We disentangle the two objectives of KT algorithms and intentionally integrate the semantic context of both questions and KCs ($\rightarrow$ limitation 2). Here, we first introduce a contrastive learning framework (Sec. 3.2) to generate representations for questions and KCs independently of student learning histories. Then, we leverage the learned representations of questions (Sec. 3.3 to model the sequential dynamics of student learning histories via existing KT algorithms. Importantly, our proposed framework is generalizable and can be combined with any existing KT model for additional performance improvements.

## 3  PROPOSED KCQRL FRAMEWORK

Our KCQRL framework has three main modules (see Fig. 2): (1) The first module takes the question content and annotates the question with a step-by-step solution and the corresponding knowledge concepts (KCs) (Sec. 3.1). (2) The second module learns the representations of questions by leveraging solution steps and KCs via a tailored contrastive learning objective (Sec. 3.2). (3) The third module integrates the learned question representations into existing KT models (Sec.3.3).

### 3.1  KNOWLEDGE CONCEPT ANNOTATION VIA LLMS (MODULE 1)

Our KCQRL leverages the complex reasoning abilities of an LLM $P_\phi(\cdot)$ and annotates the KCs of a question in a grounded manner. This is done in three steps: **(i)** the LLM generates the step-by-step solution to reveal the underlying techniques to solve the problem. **(ii)** The LLM then annotates the required KCs to solve the problem, informed by the solution steps. **(iii)** Finally, it further maps each solution step with its underlying KCs, which is particularly important for the second component of our framework. The details of these three steps are given below:[2]

---

[2]For each step, we provide the exact prompts in Appendix H.

**(i) Solution step generation:** For a question $q \in \mathbb{Q}$, our framework generates the solution steps $s_1, \ldots, s_n$ via chain-of-thought (CoT) prompting (Wei et al., 2022) of $P_\phi(.)$. Here, each $s_k$ is a coherent language sequence that serves as an intermediate step towards solving the problem $q$. To solve the problems, $s_k$ is sampled sequentially using a decoding algorithm; in our framework, we use temperature sampling, i.e., $s_k \sim P_\phi(s_k \mid \mathrm{prompt}(q), s_1, \ldots, s_{k-1})$, where $s_k$ is the $k$-th solution step of a question $q$ and $\mathrm{prompt}(q)$ is the CoT prompt for $q$. Note that CoT prompting allows us to decompose the solutions in multiple steps, which is especially relevant for questions that require complex problem-solving such as in Maths, Computer Science, Physics, Chemistry, and Medicine.

**(ii) KC annotation:** For the question $q$ from above, the LLM further generates relevant KCs $c_1, \ldots, c_m$ based on the question content of $q$ and the solutions steps $\{s_k\}_{k=1}^n$ generated from the previous step (i). Similar to previous step, the $c_j$ are sampled sequentially, i.e., $c_j \sim P_\phi(c_j \mid \mathrm{prompt}(q, \{s_k\}_{k=1}^n), c_1, \ldots, c_{j-1})$, where $c_j$ is the $j$-th generated KC of question $q$ and $\mathrm{prompt}(q, \{s_k\}_{k=1}^n)$ is the prompt for an LLM that leverages the question content and solution steps.

**(iii) Solution step → KC mapping:** Not all KCs have been practiced at each solution step of a problem. To better understand the association between each solution step and KC, our KCQRL framework further maps each solution step to its associated KCs. Specifically, $P_\phi(.)$ is presented with the question content $q$, the solution steps $\{s_k\}_{k=1}^n$, and the KCs $\{c_j\}_{j=1}^m$. Then, the LLM is asked to sequentially generate the relevant pairs of solution step and KC, i.e.,

$$(s_{\pi_s(l)}, c_{\pi_c(l)}) \sim P_\phi((s_{\pi_s(l)}, c_{\pi_c(l)}) \mid \mathrm{prompt}(q, \{s_k\}_{k=1}^n, \{c_j\}_{j=1}^m), (s_{\pi_s(1)}, c_{\pi_c(1)}), \ldots, (s_{\pi_s(l-1)}, c_{\pi_c(l-1)})),$$

with the following variables: $(s_{\pi_s(l)}, c_{\pi_c(l)})$ is the $l$-th generated pair of solution step and KC; $\pi_s(l)$ denotes the index of a solution step (from 1 to $n$) for the $l$-th generation; $\pi_c(l)$ analogously denotes the index of a KC (from 1 to $m$) for the $l$-th generation; and $\mathrm{prompt}(q, \{s_k\}_{k=1}^n, \{c_j\}_{j=1}^m)$ is the prompt that incorporates the question content, the solution steps, and the KCs.

## 3.2 Representation Learning of Questions (Module 2)

Our KCQRL framework provides a tailored contrastive learning (CL) approach to generate embeddings for questions and solution steps that are semantically aligned with their associated KCs.[3] A naïve approach would be to use a pre-trained LM, which can then generate general-purpose embeddings. However, such general-purpose embeddings lack the domain-specific focus required in education. (We provide evidence for this in our ablation studies in Sec. 5.2.) To address this, CL allows us to explicitly teach the encoder to bring question and solution step embeddings closer to the representations of their relevant KCs. After CL training, these 'enriched' embeddings are aggregated and used as input for the downstream KT model to improve the modeling capabilities of the latter.

We achieve the above objective via a carefully designed CL loss. CL has shown to be effective in representation learning of textual contents in other domains, such as information retrieval (Karpukhin et al., 2020; Khattab & Zaharia, 2020; Zhu et al., 2023), where CL was used to bring the relevant query and document embeddings closer. However, this gives rise to an important difference: In our framework, we leverage CL to bring question embeddings closer to their KC embeddings, and solution step embeddings to their KC embeddings, respectively.

To bring the embeddings of relevant texts closer, our CL loss should learn similar representations for the positive pair (e.g., the question and one of its KCs) in contrast to many negative pairs (e.g., the question and a KC from another question) from the same batch. In our setup, different questions in the same batch can be annotated with semantically similar KCs (e.g., UNDERSTANDING OF ADDITION vs. ABILITY TO PERFORM ADDITION). Constructing the negative pairs from semantically similar KCs adversely affects representation learning. This is an extensively studied problem in CL for other domains, and it is called *false negatives* (Chen et al., 2022; Huynh et al., 2022; Yang et al., 2022; Sun et al., 2023b; Byun et al., 2024). Informed by the literature, we carefully design a mechanism to pick negative pairs from the batch that avoids false negatives. For this, we consider the semantic information in annotated KCs, i.e., before the CL training, which we use to cluster the KCs that share similar semantics. Then, for a given KC in the positive pair, we discard other KCs in the same cluster when constructing the negative pairs.

Formally, after the annotation steps from the previous modules, a question $q_i$ has $N_i$ solution steps $\{s_{ik}\}_{k=1}^{N_i}$ and $M_i$ knowledge concepts $\{c_{ij}\}_{j=1}^{M_i}$. To avoid false negatives in CL, we cluster all KCs of

---

[3]For an introduction to CL, we refer to (Oord et al., 2018; Chen et al., 2020; He et al., 2020).

all questions in the corpus $\{c_{i'j'}\}_{i'=1,j'=1}^{N,M_{i'}}$ via a clustering algorithm $\mathcal{A}(\cdot)$. As a result, we have that, if $c_{ij}$ and $c_{i'j'}$ are semantically similar, then $\mathcal{A}(c_{ij}) = \mathcal{A}(c_{i'j'})$.

To train the encoder LM $E_\psi(\cdot)$ with our tailored CL objective, we first compute the embeddings of question content, solution steps, and KCs via

$$z_i^q = E_\psi(q_i), \qquad z_{ik}^s = E_\psi(s_{ik}), \qquad z_{ij}^c = E_\psi(c_{ij}). \tag{1}$$

We then leverage the embeddings to jointly learn (i) the representation of the question content and (ii) the representation of the solution steps. We describe (i) and (ii) below, as well as the training objective to learn both jointly.

**(i) Learning the representation of a question content (via $\mathcal{L}_{\mathbf{question}_i}$):** Each question in the corpus can be associated with more than one KCs. Hence, the objective here is to simultaneously achieve two objectives: (a) bring the embeddings of a given question closer to the embeddings of its KCs ($\triangleq$positive pairs) and (b) push it apart from the embeddings of irrelevant KCs ($\triangleq$negative pairs without false negatives). For a question $q_i$ and its KC $c_{ij}$, we achieve this by the following loss

$$\mathcal{L}_q(z_i^q, z_{ij}^c) = -\log \frac{\text{sim}(z_i^q, z_{ij}^c)}{\text{sim}(z_i^q, z_{ij}^c) + \sum_{\substack{i' \in \mathcal{B} \\ i' \neq i}} \sum_{j'=1}^{M_{i'}} \mathbb{I}_{\{\mathcal{A}(c_{ij}) \neq \mathcal{A}(c_{i'j'})\}} \text{sim}(z_i^q, z_{i'j'}^c)}, \tag{2}$$

where $\text{sim}(z_i^q, z_{ij}^c) = \exp\left(\frac{z_i^q \cdot z_{ij}^c}{\tau \|z_i^q\| \cdot \|z_{ij}^c\|}\right)$ and where $\mathcal{B}$ is the set of question IDs in the batch. The indicator function $\mathbb{I}_{\{\mathcal{A}(c_{ij}) \neq \mathcal{A}(c_{i'j'})\}}$ (in blue) eliminates the false negative question-KC pairs. Note that the latter distinguishes our custom CL loss from the standard CL loss.

Recall that each question $q_i$ can have multiple KCs, i.e., $\{c_{ij}\}_{j=1}^{M_i}$. Hence, we bring the embedding of $q_i$ closer to *all* of its KCs via

$$\mathcal{L}_{\text{question}_i} = \frac{1}{M_i} \sum_{j=1}^{M_i} \mathcal{L}_q(z_i^q, z_{ij}^c). \tag{3}$$

**(ii) Learning the representation of a solution step (via $\mathcal{L}_{\mathbf{step}_i}$):** We now proceed similarly to our KC annotation, which is also grounded in the solution steps. Hence, we carefully train our encoder $E_\psi(\cdot)$ in a grounded manner to the solution steps. Here, our objective is two-fold: (a) to bring the embedding of a solution step closer to the embeddings of its KCs, and (b) to separate it from the embeddings of irrelevant KCs. For a solution step $s_{ik}$ and its KC $c_{ij}$, we achieve this by the following loss

$$\mathcal{L}_s(z_{ik}^s, z_{ij}^c) = -\log \frac{\text{sim}(z_{ik}^s, z_{ij}^c)}{\text{sim}(z_{ik}^s, z_{ij}^c) + \sum_{i' \in \mathcal{B}} \sum_{j'=1}^{M_{i'}} \mathbb{I}_{\{\mathcal{A}(c_{ij}) \neq \mathcal{A}(c_{i'j'})\}} \text{sim}(z_{ik}^s, z_{i'j'}^c)}, \tag{4}$$

where $\text{sim}(\cdot, \cdot)$, $\mathcal{B}$ and the indicator function (in blue) are defined in the same as earlier.

Different from (i), not all KCs $\{c_{ij}\}_{j=1}^{M_i}$ of a question $q_i$ are relevant to a particular solution step $s_{ik}$. For this, we leverage the mapping from solution step to KC from the first module of our KCQRL framework. Based on this mapping, we define $\mathcal{P}(s_{ik})$ as the set of relevant KCs (i.e., $c_{ij}$) for $s_{ik}$, so that we can consider only the relevant KCs in the loss. Overall, for a question $q_i$, we compute the loss via

$$\mathcal{L}_{\text{step}_i} = \frac{1}{N_i} \sum_{k=1}^{N_i} \frac{1}{|\mathcal{P}(s_{ik})|} \sum_{j \in \mathcal{P}(s_{ik})} \mathcal{L}_s(z_{ik}^s, z_{ij}^c). \tag{5}$$

**Training objective:** Our framework *jointly* learns the representations of the questions and the representations of the corresponding solution steps. Specifically, for a batch of questions $\mathcal{B}$, the overall training objective is given by

$$\mathcal{L} = \frac{1}{|\mathcal{B}|} \sum_{i \in \mathcal{B}} \mathcal{L}_{\text{question}_i} + \alpha \mathcal{L}_{\text{step}_i}, \tag{6}$$

where $\alpha$ controls the balance between the contributions of $\mathcal{L}_{\text{question}_i}$ and $\mathcal{L}_{\text{step}_i}$.[4]

---

[4] Since both losses are already normalized by the number of KCs and steps, our initial experiments indicated that a choice of $\alpha = 1$ yields very good performance and that the need for additional hyperparameter tuning can be circumvented. Hence, we set $\alpha = 1$ for all experiments.

### 3.3 Improving Knowledge Tracing via Learned Question Embeddings (Module 3)

The final component of our KCQRL framework proceeds in a simple yet effective manner by integrating the learned representations of each question into existing KT models. To achieve this, we simply replace the randomly initialized question embeddings of the KT model $F_\theta$ with our learned representations. To keep the dimensionality of the embeddings consistent across the questions with different number of solution steps, we calculate the embedding of a question via

$$\text{Emb}(q_i) = [z_i^q; \tilde{z}_i^s], \qquad \tilde{z}_i^s = \frac{1}{N_i} \sum_{k=1}^{N_i} z_{ik}^s. \qquad (7)$$

After this step, the KT model $F_\theta$ is trained based on its original loss function.

Note that, without loss of generality, the above approach can be applied to KT models that capture sequences of KCs over time. Here, the KC embeddings are replaced with our learned question embeddings, and, for the training/testing, question IDs are provided instead of KC IDs. If a KT model leverages both question and KC embeddings, we then replaced the question embeddings with our learned embeddings and fix KC embeddings to a vector with zeros to have a fair comparison and to better demonstrate the effectiveness of our work.

## 4 Experimental Setup

**Datasets:** We show the effectiveness of our framework on two large-scale, real-world datasets for which high-quality question contents are available (Table 1). These are: **XES3G5M** (Liu et al., 2023c) and **Eedi** (Eedi, 2024). Both datasets are collected from online math learning platforms and are widely used to model the learning processes of students. Both datasets include data from thousands of students and questions, and millions of interactions, and, hence, these datasets are ideal for benchmarking various KT models and effectively demonstrating the impact of question semantics. Of note, the XES3G5M dataset was originally composed in Chinese, which we translated into English and made it available for future research. We additionally provide KC annotations of our KCQRL framework (i.e., the output of our module 1 in Sec. 3.1) for the XES3G5M dataset. We provide further details in Appendix B.

Table 1: Overview of datasets.

| Dataset | # Students | # KCs | # Questions | # Interactions | Language |
|---|---|---|---|---|---|
| XES3G5M (Liu et al., 2023c) | 18,066 | 865 | 7,652 | 5,549,635 | English (translated from Chinese) |
| Eedi (Eedi, 2024) | 47,560 | 1,215 | 4,019 | 2,324,162 | English |

**Baselines:** Note that our framework is highly flexible and works with any state-of-the-art KT model. We thus consider a total of 15 KT models: **DKT** (Piech et al., 2015), **DKT+** (Yeung & Yeung, 2018), **KQN** (Lee & Yeung, 2019), **qDKT** (Sonkar et al., 2020), **IEKT** (Long et al., 2021), **AT-DKT** (Liu et al., 2023a), **QIKT** (Chen et al., 2023), **DKVMN** (Zhang et al., 2017), **DeepIRT** (Yeung, 2019), **ATKT** (Guo et al., 2021), **SAKT** (Pandey & Karypis, 2019), **SAINT** (Choi et al., 2020a), **AKT** (Ghosh et al., 2020), **simpleKT** (Liu et al., 2023b). and **sparseKT** (Huang et al., 2023). For all the baselines, we make the following comparison: (a) the original implementations without our framework versus (b) the KT model together with our KCQRL framework. Hence, any performance improvement must be attributed to our framework.

We leverage pykt library (Liu et al., 2022b) for the implementation. We follow prior literature (e.g., Piech et al., 2015; Sonkar et al., 2020; Zhou et al., 2024) and evaluate the performance of KT models based on the AUC.

**Implementation details of KCQRL:** We leverage the reasoning abilities of GPT-4o[5] for our LLM $P_\phi(\cdot)$ in Sec. 3.1. We provide the exact prompts in Appendix H. We use BERT (Devlin, 2019) as our LLM encoder $E_\psi(\cdot)$ for representation learning in Sec. 3.2. For the elimination of false negative pairs, we use HDBSCAN (Campello et al., 2013) as the clustering algorithm $\mathcal{A}(\cdot)$ over Sentence-BERT (Reimers & Gurevych, 2019) embeddings of KCs. For training $E_\psi(\cdot)$, we use Nvidia Tesla A100 with 40GB GPU memory. We follow the standard five-fold cross-validation procedure to tune the parameters of the KT models $F_\theta(\cdot)$ in Sec. 3.3 and report results on the test set. For the KT models, we use NVIDIA GeForce RTX 3090 with 24GB GPU memory. Further details are given in Appendix D.

---

[5]The exact version is gpt-4o-2024-05-13 at https://platform.openai.com/docs/models/gpt-4o

# 5 RESULTS

## 5.1 PREDICTION PERFORMANCE

**Main results:** Table 2 reports the performance of different KT models, where we each compare in two variants: (a) without (i.e., Default) versus (b) with our framework. We find the following: **(1)** Our KCQRL framework consistently boosts performance across all KT models and across all datasets. The relative improvements range between 1% to 7%. **(2)** Our framework improves the state-of-the-art (SOTA) performance on both datasets. For XES3G5M, the AUC increases from 82.24 to 83.04 (**+0.80**), and, for Eedi, from 75.15 to 78.96 (**+3.81**). **(3)** For KT models that have otherwise a lower performance, the performance gains from our KCQRL framework are particularly large. Interestingly, this helps even low-performing KT models to reach near-SOTA performance when provided with semantically rich inputs from our framework. *Takeaway:* Our framework leads to consistent performance gains and achieves SOTA performance.

Table 2: **Improvement in the performance of KT models from our KCQRL framework.** Shown: AUC with std. dev. across 5 folds. Improvements are shown as both absolute and relative (%) values.

| Model | XES3G5M | | | | Eedi | | | |
|---|---|---|---|---|---|---|---|---|
| | **Default** | **w/ KCQRL** | **Imp. (abs.)** | **Imp. (%)** | **Default** | **w/ KCQRL** | **Imp. (abs.)** | **Imp. (%)** |
| DKT | 78.33 ± 0.06 | 82.13 ± 0.02 | +3.80 | +4.85 | 73.59 ± 0.01 | 74.97 ± 0.03 | +1.38 | +1.88 |
| DKT+ | 78.57 ± 0.05 | 82.34 ± 0.04 | +3.77 | +4.80 | 73.79 ± 0.03 | 75.32 ± 0.04 | +1.53 | +2.07 |
| KQN | 77.81 ± 0.03 | 82.10 ± 0.06 | +4.29 | +5.51 | 73.13 ± 0.01 | 75.16 ± 0.04 | +2.03 | +2.78 |
| qDKT | 81.94 ± 0.05 | 82.13 ± 0.05 | +0.19 | +0.23 | 74.09 ± 0.03 | 74.97 ± 0.04 | +0.88 | +1.19 |
| IEKT | **82.24 ± 0.07** | 82.82 ± 0.06 | +0.58 | +0.71 | 75.12 ± 0.02 | 75.56 ± 0.02 | +0.44 | +0.59 |
| AT-DKT | 78.36 ± 0.06 | 82.36 ± 0.07 | +4.00 | +5.10 | 73.72 ± 0.04 | 75.25 ± 0.02 | +1.53 | +2.08 |
| QIKT | 82.07 ± 0.04 | 82.62 ± 0.05 | +0.55 | +0.67 | **75.15 ± 0.04** | 75.74 ± 0.02 | +0.59 | +0.79 |
| DKVMN | 77.88 ± 0.04 | 82.64 ± 0.02 | +4.76 | +6.11 | 72.74 ± 0.05 | 75.51 ± 0.02 | +2.77 | +3.81 |
| DeepIRT | 77.81 ± 0.06 | 82.56 ± 0.02 | +4.75 | +6.10 | 72.61 ± 0.02 | 75.18 ± 0.05 | +2.57 | +3.54 |
| ATKT | 79.78 ± 0.07 | 82.37 ± 0.04 | +2.59 | +3.25 | 72.17 ± 0.03 | 75.28 ± 0.04 | +3.11 | +4.31 |
| SAKT | 75.90 ± 0.05 | 81.64 ± 0.03 | +5.74 | +7.56 | 71.60 ± 0.03 | 74.77 ± 0.02 | +3.17 | +4.43 |
| SAINT | 79.65 ± 0.02 | 81.50 ± 0.07 | +1.85 | +2.32 | 73.96 ± 0.02 | 75.20 ± 0.04 | +1.24 | +1.68 |
| AKT | 81.67 ± 0.03 | **83.04 ± 0.05** | +1.37 | +1.68 | 74.27 ± 0.03 | 75.49 ± 0.03 | +1.22 | +1.64 |
| simpleKT | 81.05 ± 0.06 | 82.92 ± 0.04 | +1.87 | +2.31 | 73.90 ± 0.04 | 75.46 ± 0.02 | +1.56 | +2.11 |
| sparseKT | 79.65 ± 0.11 | 82.95 ± 0.09 | +3.30 | +4.14 | 74.98 ± 0.09 | **78.96 ± 0.08** | +3.98 | +5.31 |

Best values are in bold. The shading in green shows the magnitude of the performance gain.

**Sensitivity to the number of students:** Fig. 3 shows the prediction results for each KT model under a varying number of students available for training (starting from 5% of the actual number of students). Our KCQRL significantly improves the performance of KT models for both datasets and for all % of available training data. In general, the performance gain from our framework tends to be larger for small-student settings. For instance, the performance of DKVMN on Eedi improves by almost +6 AUC even when *only* 5% of the actual number of students are available for training. This underscores the benefits of our KCQRL during the early phases of online learning platforms and for specialized platforms (e.g., in-house training for companies). The results for other KT models are given in Appendix E. *Takeaway:* Our framework greatly helps generalization performance, especially in low-sample size settings.

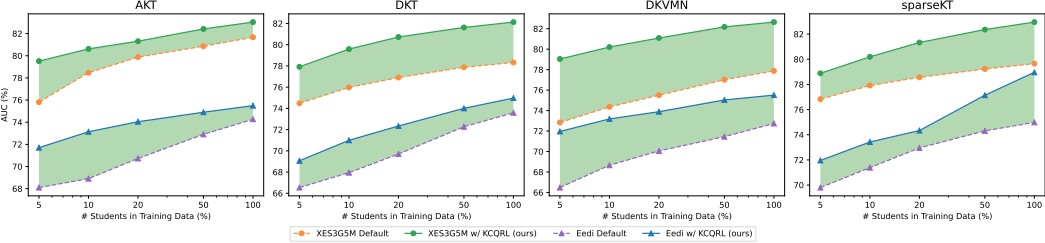

Figure 3: **Improvement of our KCQRL across different training set sizes**. Plots show different KT models, where, on the x-axis, we report the performance when varying the number of students in our datasets. Green area covers the improvement from our framework.

**Multi-step ahead prediction:** We now follow Liu et al. (2022b) and focus on the task of predicting a span of student's responses given the history of interactions. Example: given the initial 60% of the interactions, the task is to predict the student's performance in the next 40% of the exercises. We distinguish two scenarios (Liu et al., 2022b): (a) accumulative or (b) non-accumulative manner,

where (a) means that KT models makes predictions in a rolling manner (so that predictions for the $n$-outcome are used to prediction the $(n + 1)$-th) and (b) means that the KT models predict all future values at once (but assume that the predictions are independent).

Fig. 4 presents the results. Overall, our KCQRL improves the performance of KT models for both datasets and for both settings. The only exception is sparseKT during the early stages of accumulative prediction on Eedi. Detailed results for other KT models are given in Appendix F. *Takeaway:* Our framework greatly improves long-term predictions of students' learning journeys and their outcomes.

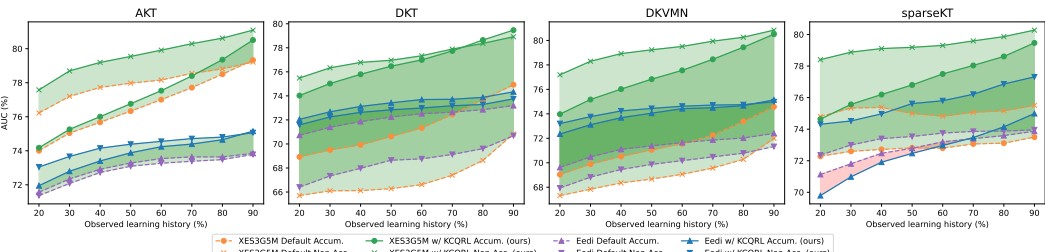

Figure 4: **Improvement of our KCQRL in multi-step-ahead prediction.** Plots show different KT models, where we vary the portion of observed learning history and predict the rest of the entire learning journey. Green [red] area shows the improvement [decline] from using our framework.

## 5.2 ABLATION STUDIES

**Quality of the automated KC annotation:** Here, we perform an ablation for module 1 of our framework (in Sec. 3.1) where we assess the quality of the KC annotations. We compare against two ablations: **(a)** the original KC annotations from the dataset and **(b)** the annotations from our KCQRL but without leveraging the solution steps. For the ablation study, we ran our automated evaluation via a different LLM, i.e., Llama-3.1-405B (Dubey et al., 2024), to avoid the potential bias from using the same model as in our KC annotation.

Table 3 shows the pairwise comparison of three KC annotations (i. e., ours and two ablations) based on 5 criteria, where the LLM model is prompted to choose the best of the given two annotations for each criterion. The two key observations are: **(1)** Both annotations from KCQRL (i.e., with and without solution steps) are clearly preferred over the annotations from the original dataset, confirming the findings from Moore et al. (2024). **(2)** Overall, KC annotations from our full framework are preferred over the annotations without solution steps, confirming our motivation of our KC annotation. Further details and example KC annotations are in Appendix G. *Takeaway:* A large performance gain is due to module 1 where we ground the automated KC annotations in chain-of-thought reasoning.

Appendix J compares KC annotation quality across LLM sizes, while Appendix K presents human evaluation results comparing LLM annotations to the original dataset. Both confirm our design.

Table 3: **Ablation study showing the relevance of automated KC annotation.** We report the quality (in %) for different KC annotations.

| Criteria | XES3G5M | | | | | | Eedi | | | | | |
| | Original | KCQRL w/o sol. steps | Original | KCQRL | KCQRL w/o sol. steps | KCQRL | Original | KCQRL w/o sol. steps | Original | KCQRL | KCQRL w/o sol. steps | KCQRL |
|---|---|---|---|---|---|---|---|---|---|---|---|---|
| Correctness | 33.9 | **66.1** | 6.8 | **93.2** | 15.9 | **84.1** | 44.2 | **55.8** | 25.9 | **74.1** | 27.0 | **73.0** |
| Coverage | 41.9 | **58.1** | 13.5 | **86.5** | 13.3 | **86.7** | 25.9 | **74.1** | 7.7 | **92.3** | 22.5 | **77.5** |
| Specificity | 33.5 | **66.5** | 25.5 | **74.5** | 36.0 | **64.0** | 37.0 | **63.0** | 39.2 | **60.8** | 55.8 | 44.2 |
| Ability of Integration | 40.3 | **59.7** | 12.7 | **87.3** | 12.5 | **87.5** | 34.7 | **65.3** | 20.6 | **79.4** | 25.0 | **75.0** |
| Overall | 38.6 | **61.4** | 7.8 | **92.2** | 13.1 | **86.9** | 36.7 | **63.3** | 21.2 | **78.8** | 24.1 | **75.9** |

**Performance of question embeddings w/o CL:** We now demonstrate the effectiveness of module 2 (defined Sec. 3.2), which is responsible for the representation learning of questions. We thus compare our framework's embeddings against five ablations without any CL training from the same LLM encoder. These are the embeddings of **(a)** question content, **(b)** question and its KCs, **(c)** question and its solution steps, **(d)** question and its solution steps and KCs, and **(e)** only the KCs of a question.

Fig. 5 plots the embeddings of questions. For all five baselines, the representations of questions from the same KC are scattered very broadly. In comparison, our KCQRL is effective in bundling the questions from the same KC in close proximity as desired. *Takeaway:* This highlights the importance of our CL training in Sec. 3.2.

Fig. 6 shows the performance of KT models with these five baseline embeddings in comparison to the default implementation and our complete framework. In most cases, baseline embeddings perform better than the default implementation of KT models, which again highlights the importance of semantic representations (as compared to randomly initialized embeddings). Further, our KCQRL is consistently better than the baseline embeddings, which highlights the benefit of domain-specific representation learning. *Takeaway:* A large performance gain is from module 2, which allows our framework to capture the semantics of both questions and KCs.

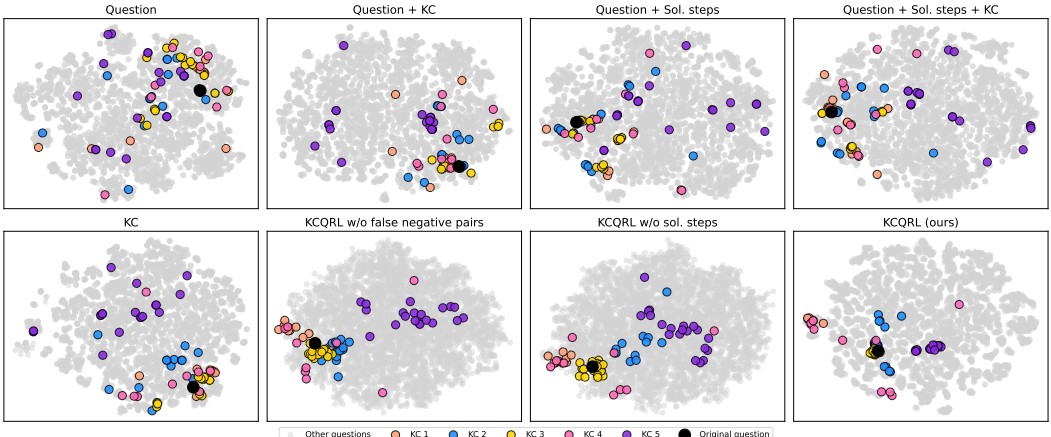

Figure 5: **Visualization of question embeddings.** For better intuition, we chose the same question as in the example from Fig. 2, and, for each of its KCs, we color the question representations sharing the same KC. Evidently, our CL loss is highly effective.

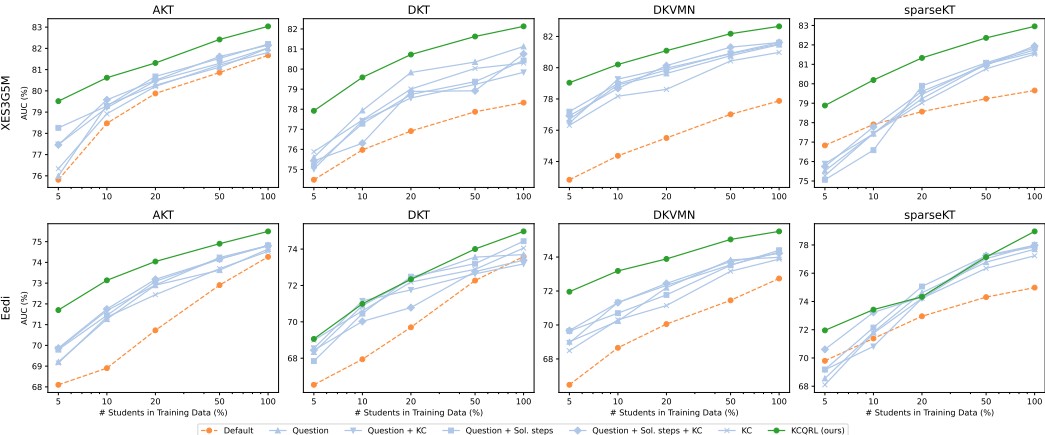

Figure 6: **Ablation studies showing the effectiveness of our representation learning.** Evaluation is done across different training set sizes.

**Ablations for CL training:** To demonstrate the effectiveness of our CL loss, we implement the following ablations: **(f)** KCQRL w/o false negative elimination, whose CL loss is calculated without the blue indicator function in Eq. 2 and Eq. 4, and **(g)** KCQRL w/o sol. steps, whose CL loss is calculated based only on $\mathcal{L}_{\text{question}_i}$ by ignoring $\mathcal{L}_{\text{step}_i}$.

Fig. 5 shows the embeddings are better organized than the ablations with no CL training. Yet, our complete framework brings the representations of questions from the same KCs much closer together.

Fig. 7 compares the different variants of the CL loss. Here, the performance improvement from using either false negative elimination or using the representation solution steps is somewhat similar (compared to a default CL loss), but the combination of both is best. *Takeaway:* Our custom CL loss outperforms the standard CL loss by a clear margin.

## 6 RELATED WORK

We provide a brief summary of relevant works below. An extended related work (including representation learning and custom CL) is in Appendix A.

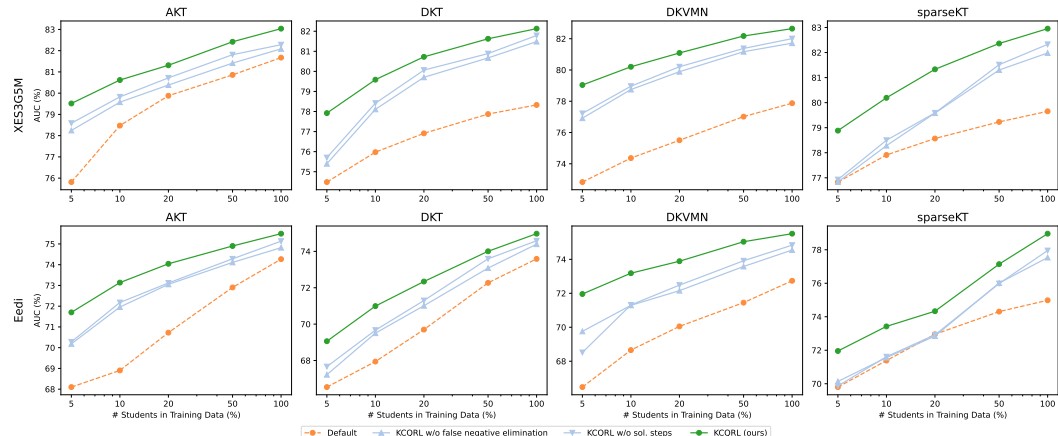

Figure 7: **Ablation studies showing the effectiveness of custom CL loss.** Evaluation is done across different training set sizes. Here, we assess the contributions of (i) false negative elimination and (ii) representation learning of solution steps to the overall performance.

**Knowledge tracing (KT)** aims to model the temporal dynamics of students' interactions with learning content to predict their knowledge over time (Corbett & Anderson, 1994). A large number of machine learning models, including deep neural networks, have been proposed for this purpose (Piech et al., 2015; Yeung & Yeung, 2018; Lee & Yeung, 2019; Sonkar et al., 2020; Long et al., 2021; Liu et al., 2023a; Chen et al., 2023; Zhang et al., 2017; Yeung, 2019; Guo et al., 2021; Pandey & Karypis, 2019; Choi et al., 2020a; Ghosh et al., 2020; Liu et al., 2023b; Huang et al., 2023).

We later use all of the above-mentioned KT models in our experiments, as our KCQRL framework is designed to enhance the performance of *any* KT model. However, existing works (1) require a comprehensive mapping between knowledge concepts and questions, and (2) overlook the semantics of question content and KCs. Our KCQRL effectively addresses both limitations, which leads to performance improvements across all of the state-of-the-art KT models.

**KC annotation** aims to infer the KCs of a given exercise in an automated manner. (i) One research line infers KCs from students' learning histories as latent states (e.g., Barnes, 2005; Cen et al., 2006; Liu et al., 2019; Shi et al., 2023). However, these latent states lack interpretability, limiting their use in KT. (ii) Another approach reformulates KC annotation as a classification task (e.g., Patikorn et al., 2019; Tian et al., 2022; Li et al., 2024a;b), but this requires large annotated datasets, making it often infeasible. (iii) Recent works have also explored KC annotation in free-text form (Moore et al., 2024; Didolkar et al., 2024), but their annotations are not informed by the question's solution steps, which we show is suboptimal in our ablations. In our work, we leverage LLMs' reasoning abilities to generate step-by-step solutions and produce grounded KCs for each question.

## 7 DISCUSSION

We present a novel framework for improving the effectiveness of any existing KT model: *automated **k**nowledge **c**oncept annotation and **q**uestion **r**epresentation **l**earning*, which we call **KCQRL**. Our KCQRL framework is carefully designed to address two key limitations of existing KT models. **Limitation 1:** Existing KT models require a comprehensive mapping between knowledge concepts (KCs) and questions, typically done manually by domain experts. **Contribution 1:** We circumvent such a need by introducing a novel, automated KC annotation module grounded in solution steps. **Limitation 2:** Existing KT models overlook the semantics of both questions and KCs, treating them as identifiers whose embeddings are learned from scratch. **Contribution 2:** We propose a novel contrastive learning paradigm to effectively learn representations of questions and KC, also grounded in solution steps. We integrate the learned representations into KT models to improve their performance. **Conclusion:** We demonstrate the effectiveness of our KCQRL across 15 KT models on two large real-world Math learning datasets, where we achieve consistent performance improvements.

**Implications:** Our KCQRL allows even the simplest KT models to reach near state-of-the-art (SoTA) performance, suggesting that much of the existing literature may have "overfit" to the paradigm of modeling sequences based only on IDs. Future KT research can advance by focusing on sequential models "designed to" inherently understand semantics. As a first step, we release an English version of the XES3G5M dataset with full annotations, including question content, solutions, and KCs.

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

## A  RELATED WORK (EXTENDED)

**Knowledge tracing (KT)** aims to model the temporal dynamics of students' interactions with learning content to predict their knowledge over time (Corbett & Anderson, 1994). Early works in KT primarily leveraged logistic and probabilistic models (Corbett & Anderson, 1994; Cen et al., 2006; Pavlik et al., 2009; Käser et al., 2017; Vie & Kashima, 2019). Later, many KT works focused on deep learning-based approaches, which can loosely be grouped into different categories: 1) Deep sequential models that leverage auto-regressive architectures such as RNN or LSTM (Piech et al., 2015; Yeung & Yeung, 2018; Lee & Yeung, 2019; Liu et al., 2019; Nagatani et al., 2019; Sonkar et al., 2020; Guo et al., 2021; Long et al., 2021; Shen et al., 2022; Chen et al., 2023; Liu et al., 2023a), 2) Memory augmented models to externally capture the latent states of the students during the learning process (Zhang et al., 2017; Abdelrahman & Wang, 2019; Yeung, 2019), 3) Graph-based models that use graph neural networks to capture the interactions between knowledge concepts and questions (Nakagawa et al., 2019; Song et al., 2022; Cui et al., 2024), 4) Attention-based models that either employ a simple attention mechanism or transformer-based encoder-decoder architecture for the modeling of student interactions (Pandey & Karypis, 2019; Pandey & Srivastava, 2020; Choi et al., 2020a; Ghosh et al., 2020; Huang et al., 2023; Im et al., 2023; Liu et al., 2023b; Yin et al., 2023; Ke et al., 2024; Li et al., 2024c). Outside these categories, Lee et al. (2024) fits entire history of students exercises into the context window of a language model to make the prediction, Guo et al. (2021) designs an adversarial training to have robust representations of latent states, and Zhou et al. (2024) develops a generative model to track the knowledge states of the students. However, the existing works (1) require a comprehensive mapping between knowledge concepts and questions and (2) overlook the semantics of questions' content and KCs. Our KCQRL framework effectively addresses both limitations.

**KC annotation** aims at inferring the KCs of a given exercise in an automated manner. (i) One line of research infers KCs of questions by learning patterns from students' learning histories in the form of latent states (e.g., Barnes, 2005; Cen et al., 2006; Liu et al., 2019; Shi et al., 2023). Yet, such latent states lack interpretability, because of which their use in KT is typically prohibited. (ii) Another line of research reformulates KCs annotation as a classification task, typically via model training (Patikorn et al., 2019; Tian et al., 2022) or LLM prompting (Li et al., 2024a;b). Although these works do not require human experts at inference, they still require them to curate a large annotated dataset for models to learn the task. (iii) Recent works also explored the KC annotation in free-text form (Moore et al., 2024; Didolkar et al., 2024). However, their KC annotations are not informed by the solution steps of the question. We show that this is suboptimal in our ablations.

In our work, we leverage the reasoning abilities of LLMs (Wei et al., 2022; Fu et al., 2023; Shridhar et al., 2023; Wang et al., 2023; Yu et al., 2024) to generate the step-by-step solution steps of a given question and generate the associated KCs in a grounded manner.

**Representation learning** for textual data has been extensively studied in the context of LLMs. While pre-trained LLMs generate effective general-purpose semantic representations, these embeddings often turn out to be suboptimal for domain-specific tasks (Karpukhin et al., 2020). Through CL training, the model learns to bring the positive pairs of textual data closer together (but not the negative pairs in the embeddings).

Many works designed tailored CL losses (Oord et al., 2018; Chen et al., 2020; He et al., 2020; Ozyurt et al., 2023b) to improve the representations of textual data in various domains. Examples are information retrieval (Karpukhin et al., 2020; Khattab & Zaharia, 2020; Lee et al., 2023; Sun et al., 2023a; Zhu et al., 2023; Zhao et al., 2024), textual entailment (Gao et al., 2021; Li & Li, 2024; Ou & Xu, 2024), code representation learning (Zhang et al., 2024), and question answering (Yue et al., 2021; 2022). To our knowledge, we are the first to leverage a CL loss for the representation learning of questions specific to KT tasks.

The prevalence of *false negative* pairs can significantly degrade the quality of CL (Saunshi et al., 2019), as it causes semantically similar samples to be incorrectly pushed apart. To address this, several works in computer vision have proposed methods to detect and eliminate false negative pairs by leveraging the similarity of sample embeddings during training (Chen et al., 2022; Huynh et al., 2022; Yang et al., 2022; Sun et al., 2023b; Byun et al., 2024). Different from these works, we develop a custom CL approach that is carefully designed to our task. Therein, we detect similarities between samples (i.e., KCs) in advance (e. g., UNDERSTANDING OF ADDITION vs. ABILITY TO PERFORM

ADDITION), and eliminate false negative question-KC and solution step-KC pairs during our custom CL training.

# B DATASET DETAILS

**XES3G5M** (Liu et al., 2023c): XES3G5M contains the history student exercises for 7,652 unique questions. These questions are mapped to 865 unique KCs in total. It has a total of 18,066 students' learning histories and a total of 5,549,635 interactions for all students. Fig. 8a demonstrates the distribution of interaction numbers across students. Of note, XES3G5M dataset is original provided in Chinese, which we translated to English to make it compatible to our framework. Details are given below.

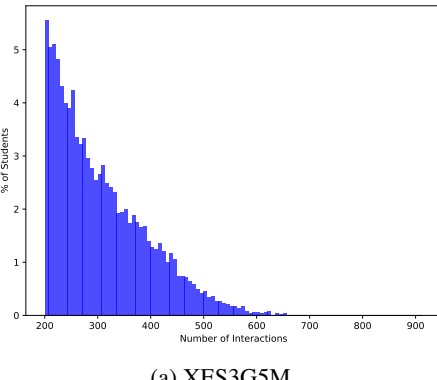

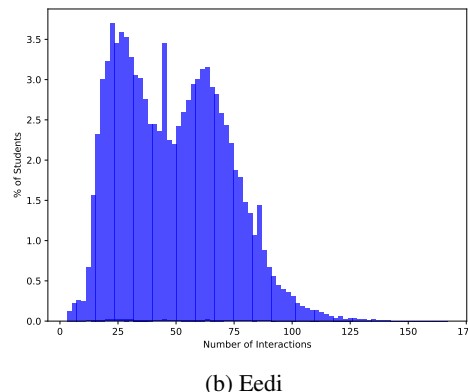

(a) XES3G5M

(b) Eedi

Figure 8: Histogram of number of interactions for each dataset.

**Translation:** The original XES3G5M dataset is provided in Chinese. To make it compatible with our KCQRL framework, we first translated the question contents to English. We did the translation by Google Translate via deep-translate Python library.

**Conversion to proper question format:** XES3G5M dataset contains 6,142 fill-in-the-blank style questions out of 7,652. After the translation, we manually inspected the quality of question contents and found that blanks are disappeared in the translation for fill-in-the-blank questions. For instance, one question is translated as "...There are different ways to wear it.", which should have been "...There are ___ different ways to wear it."

To make the fill-in-the-blank questions consistent with the others (and also consistent with our other dataset Eedi), We prompt GPT-4o to convert these questions to proper question phrases. From the same example, "...There are ___ different ways to wear it." is converted to "...How many ways are there to wear it?". We provide our prompt template in Appendix B.1.

**Eedi** (Eedi, 2024): Eedi contains the history student exercises for 4,019 unique questions. These questions are mapped to 1215 unique KCs in total. It has a total of 47,560 students' learning histories and a total of 2,324,162 interactions for all students. Fig. 8b demonstrates the distribution of interaction numbers across students.

### B.1 PROMPT FOR CONVERTING XES3G5M TO PROPER QUESTION FORMAT

Below we show our prompt to convert the fill-in-the-balnk style questions into proper question phrases.

---

**Prompt**

You have narrative-based math problems that already contain all the necessary information, including what needs to be solved. Your goal is to write "Converted" field by minimally modifying the last part of the "Original" field to turn them into explicit questions. This should be done in such a way that it retains the original content and context, only slightly altering the phrasing to form a question.

Example:
Original: During the Spring Festival, Xue Xue and her parents went back to their hometown to visit their grandparents. They had to take a long-distance bus for 2 hours. The speed of the long-distance bus was 85 kilometers per hour. So, Xue Xue walked a total of one kilometer from home to my grandparents' house.
Converted: During the Spring Festival, Xue Xue and her parents went back to their hometown to visit their grandparents. They had to take a long-distance bus for 2 hours. The speed of the long-distance bus was 85 kilometers per hour. So, how many kilometers did Xue Xue travel in total from home to her grandparents' house?

Now, convert the "Original" field into a question and write it into "Converted" field.
Original: **<CHINESE QUESTION CONTENT>**
Converted:

---

## C    KT Models Details

In our KCQRL framework, we enhanced the performance of 15 KT algorithms on two real-world large online math learning datasets. The summary of these KT algorithms are given below:

- **DKT** (Piech et al., 2015): It is the first deep learning based KT algorithm. It uses the LSTM to model the temporality in students' learning histories. The original DKT turns each KC into a one-hot or a random vector. The recent implementation from pykt (Liu et al., 2022b) learns the embeddings for each KC, which are then processed by an LSTM layer.

- **DKT+** (Yeung & Yeung, 2018): It adds regularization to the existing DKT model. Specifically, it adds a reconstruction loss to the last exercise's prediction. Further, it adds a regularization term to make the predictions of the same KC consistent across the time dimension.

- **KQN** (Lee & Yeung, 2019): It models the students' learning histories via RNN. Further, it has an explicit neural network mechanism to capture the latent knowledge states.

- **qDKT** (Sonkar et al., 2020):It is a variant of DKT that learns the temporal dynamics of students' learning processes via the questions rather than KCs.

- **IEKT** (Long et al., 2021): It models the students' learning histories via RNN. In addition, it has two additional neural network modules, namely student cognition and knowledge acquisition estimation.

- **AT-DKT** (Liu et al., 2023a): It has the same model backbone as in DKT. On top of it, AT-DKT has two auxiliary tasks, question tagging prediction and student's individual prior knowledge prediction.

- **QIKT** (Chen et al., 2023): It combines three neural network modules to model the learning processes, question-centric knowledge acquisition module via LSTM, question-agnostic knowledge state module via another LSTM and question centric problem solving module via MLP.

- **DKVMN** (Zhang et al., 2017): It employs key-value memory networks to model the relationships between the latent concepts and output the student's knowledge mastery of each concept.

- **DeepIRT** (Yeung, 2019): It incorporates the architecture of DKVMN and further leverages the item-response theory (Rasch, 1993) to make interpretable predictions.

- **ATKT** (Guo et al., 2021): It models the students learning histories via LSTM. Further, it employs an adversarial training mechanism to increase the generalization of model predictions.

- **SAKT** (Pandey & Karypis, 2019): It develops a simple self-attention mechanism to model the learning histories and make the prediction.

- **SAINT** (Choi et al., 2020a): It employs a Transformer-based model that is using mulitple layers of encoders and decoders to model the history of exercise information and responses.

- **AKT** (Ghosh et al., 2020): This model employs a monotonic attention mechanism between the exercises via exponential decay over time. It also employs Rasch model (Rasch, 1993) to characterize the question's difficulty and the learner's ability.

- **simpleKT** (Liu et al., 2023b): As a simple but tought to beat baseline, this model employs and ordinary dot product as an attention mechanism to the embeddings of questions.

- **sparseKT** (Huang et al., 2023): On top of simpleKT, this model further introduces a sparse attention mechanism such that for the next exercise's prediction, the model attends to at most $K$ exercises in the past.

The implementation details of the KT models are given in Appendix D.

# D IMPLEMENTATION DETAILS

Here we explain the implementation details of each part of our KCQRL framework.

**KC Annotation (Sec. 3.1):** We leverage the reasoning abilities of OpenAI's GPT-4o[6] model as our LLM $P_\phi(\cdot)$ at each step. We set its temperature parameter to 0 to get the deterministic answers from the model. For each question in the dataset, GPT-4o model is prompted three times in total: one for solution steps generation, one for KC annotation and one for solution step-KC mapping. Overall, the cost is around 80 USD for XES3G5M dataset with 7,652 questions and 50 USD for Eedi dataset with 4,019 questions.

**Representation learning of questions Sec. 3.2:** We train BERT (Devlin, 2019) as our LLM encoder $E_\psi(\cdot)$. For the elimination of false negative pairs, we use HDBSCAN (Campello et al., 2013) as the clustering algorithm $\mathcal{A}(\cdot)$. The clustering is done over the Sentence-BERT (Reimers & Gurevych, 2019) embeddings of KCs, which is a common practice in identifying relevant textual documents (Liu et al., 2022a; Ozyurt et al., 2023a). For HDBSCAN clustering, we set minimum cluster size to 2, minimum samples to 2, metric to cosine similarity between the embeddings. For contrastive learning training, we train BERT (Devlin, 2019) as our LLM encoder. To better distinguish three types of inputs, i. e. question content, solution step, and KC, we introduce three new tokens [Q], [S], [KC] to be learned during the training. These new tokens are added to the beginning of question content, solution step, and KC, respectively, for both training and inference. For all input types, we use [CLS] token's embeddings as the embeddings of the entire input text. For the training objective, we set $\alpha = 1$. Since both losses $\mathcal{L}_{\text{question}_i}$ and $\mathcal{L}_{\text{step}_i}$ are already normalized by the number of KCs and steps, our initial exploration ($\alpha$ varying from 0.2 to 5.0) indicated that setting $\alpha = 1$ yields the best performance. We train the encoder for 50 epochs with the following hyperparameters: batch size = 32, learning rate = 5e-5, dropout = 0.1, and temperature (of similarity function) = 0.1. For training of $E_\psi(\cdot)$, we use Nvidia Tesla A100 with 40GB GPU memory. The entire training is completed under 4 hours.

After the training, the question embeddings are acquired by running the inference on question text and solution step texts for each question in the dataset. Specifically, we take the embeddings of "[Q] <question content>" for the question content and "[S] <solution step>" for each solution step. Then, we aggregate these embeddings as explained in Sec. 3.3. As a result, the inference does not require the KC annotations. This has the following advantage: When a new question is added to the dataset, the earlier KC annotation module can be skipped completely and one can directly get the embeddings in this module and start using them for the downstream KT model.

**Improving KT via learned question embeddings:** We adopt the following strategy for the fair evaluation of KT algorithms and their improvements via our novel KCQRL framework. We first did a grid search over the hyperparameters of each KT model (see Table 4 for parameters) to find the best configuration for each model. Then, we replaced their embeddings with our learned question embeddings and trained/tested the KT models with the same configurations found earlier (i. e., no grid search is applied). To ensure the fair evaluation between the KT algorithms, and between their default versions and improved versions (via our framework), we fixed their embedding dimensions to 300. As BERT embeddings' default dimensionality is larger (i. e., 768), we just added a linear layer (no non-linear activation function is added) on top of the replaced embeddings to reduce its dimensionality to 300 to ensure that model capacities are subjected to a fair comparison. As KT models are much smaller than the LLM encoders, this time we used NVIDIA GeForce RTX 3090 with 24GB GPU memory for even larger batch sizes.

**Evaluation of KC-centric KT models:** Some KT models (such as DKT (Piech et al., 2015), simpleKT (Liu et al., 2023b) etc.) expand the sequence of questions into the sequence of KCs for both training and inference. For instance, if there are 3 questions with 2 KCs each, they transform the sequence $\{q_1, q_2, q_3\}$ into $\{c_{11}, c_{12}, c_{21}, c_{22}, c_{31}, c_{32}\}$ and assign the labels of original questions to each of their corresponding KCs for training. This paradigm causes information leakage in the evaluation as highlighted by Liu et al. (2022b). The reason is, during the prediction of $c_{32}$, the label of $c_{31}$ is already given as the history, which is the same label (to be predicted) for $c_{32}$. To eliminate the information leakage, we followed the literature (Liu et al., 2022b; 2023c) and applied the following procedure. Again from the same example, to predict the label of $q_3$, 1) we provided the expanded KC

---

[6]https://platform.openai.com/docs/models/gpt-4o

sequence of earlier questions as before, i. e., $\{c_{11}, c_{12}, c_{31}, c_{32}\}$. 2) Then we appended $c_{31}$ and $c_{32}$ separately to the given sequence and run the predictions independently. 3) Finally we aggregated these predictions by taking their mean, and used it as the model's final prediction. **Important note:** With our KCQRL framework, these KT models do not suffer the information leakage anymore, as all models learn from the sequence of questions with our improved version.

**Scalability:** Our novel KCQRL framework scales well to any KT model with no additional computational overhead. Specifically, our KC annotation and representation learning modules are completed independently from the KT models, and they need to run only one time in the beginning. After the embeddings are computed, they can be used in any standard KT model without impacting the models' original runtimes.

Table 4: Hyperparameter tuning of KT algorithms.

| Method | Hyperparameter | Tuning Range |
|---|---|---|
| All methods | Embedding size | 300 |
| | Batch size | 32, 64, 128 |
| | Dropout | 0, 0.1, 0.2 |
| | Learning rate | $[1 \cdot 10^{-4}, 1 \cdot 10^{-3}]$ |
| DKT | LSTM hidden dim. | 64, 128 |
| DKT+ | $\lambda_r$ | 0.005, 0.01, 0.02 |
| | $\lambda_{w_1}$ | 0.001, 0.003, 0.05 |
| | $\lambda_{w_2}$ | 1, 3, 5 |
| KQN | # RNN layers | 1, 2 |
| | RNN hidden dim. | 64, 128 |
| | MLP hidden dim. | 64, 128 |
| qDKT | LSTM hidden dim. | 64, 128 |
| IEKT | $\lambda$ | 10, 40, 100 |
| | # Cognitive levels | 5, 10, 20 |
| | # Knowledge Acquisition levels | 5, 10, 20 |
| AT-DKT | $\lambda_{pred}$ | 0.1, 0.5, 1 |
| | $\lambda_{his}$ | 0.1, 0.5, 1 |
| QIKT | $\lambda_{q_{all}}$ | 0, 0.5, 1 |
| | $\lambda_{c_{all}}$ | 0, 0.5, 1 |
| | $\lambda_{q_{next}}$ | 0, 0.5, 1 |
| | $\lambda_{c_{next}}$ | 0, 0.5, 1 |
| DKVMN | # Latent state | 10, 20, 50, 100 |
| DeepIRT | # Latent state | 10, 20, 50, 100 |
| ATKT | Attention dim. | 64, 128, 256 |
| | $\epsilon$ | 5, 10, 20 |
| | $\beta$ | 0.1, 0.2, 0.5 |
| SAKT | Attention dim. | 64, 128, 256 |
| | # Attention heads | 4, 8 |
| | # Encoders | 1, 2 |
| SAINT | Attention dim. | 64, 128, 256 |
| | # Attention heads | 4, 8 |
| | # Encoders | 1, 2 |
| AKT | Attention dim. | 64, 128, 256 |
| | # Attention heads | 4, 8 |
| | # Encoders | 1, 2 |
| simpleKT | Attention dim. | 64, 128, 256 |
| | # Attention heads | 4, 8 |
| | $L_1$ | 0.2, 0.5, 1 |
| | $L_2$ | 0.2, 0.5, 1 |
| | $L_3$ | 0.2, 0.5, 1 |
| sparseKT | Attention dim. | 64, 128, 256 |
| | # Attention heads | 4, 8 |
| | $L_1$ | 0.2, 0.5, 1 |
| | $L_2$ | 0.2, 0.5, 1 |
| | $L_3$ | 0.2, 0.5, 1 |
| | Top K | 5, 10, 20 |

Other than specified parameters, we use the default values from pykt library.

# E  SENSITIVITY TO THE NUMBER OF STUDENTS

Fig. 9 demonstrates the extended prediction results for KT models with varying numbers of students available for training (starting from 5 % of the total number of students). Across all 15 KT models, our KCQRL consistently improves performance on both datasets and for all ranges of student numbers. Overall, our framework significantly enhances generalization, especially in low-sample settings. This highlights the advantages of KCQRL in the early phases of online learning platforms, where the number of students using the system is limited.

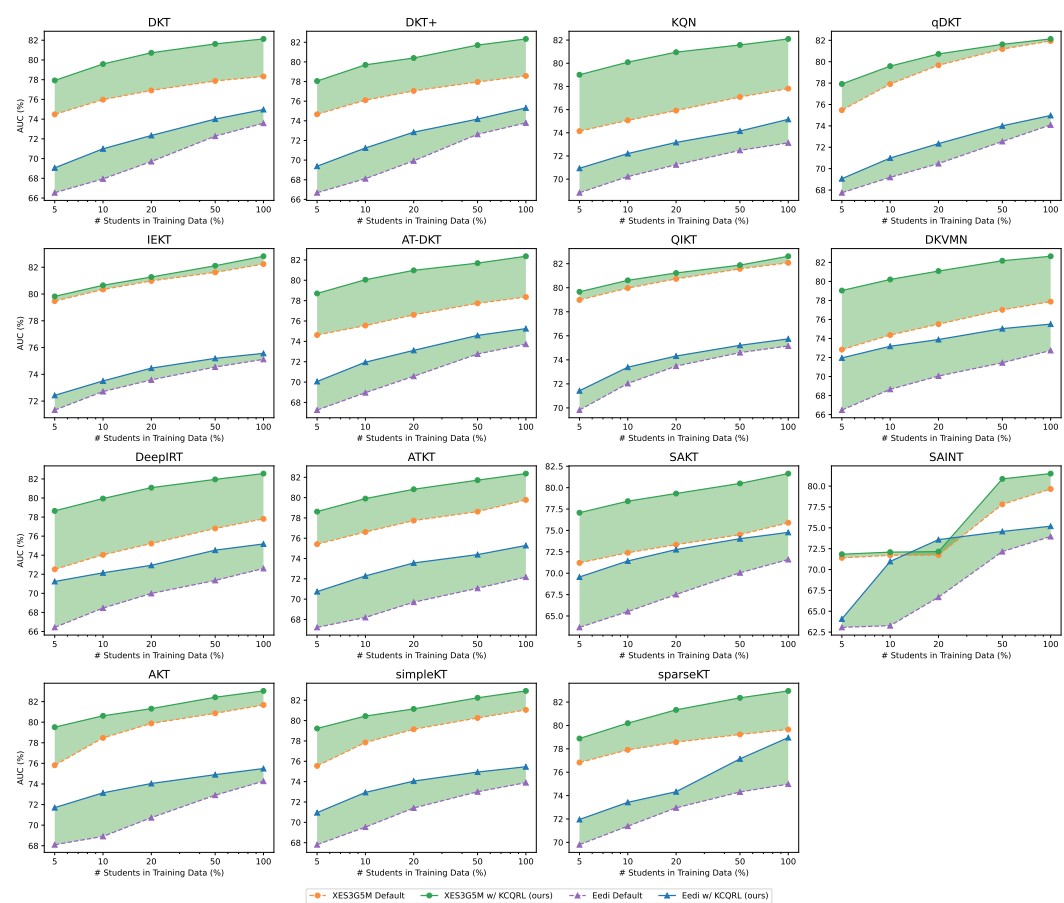

Figure 9: Improvement of our KCQRL across models and datasets with varying availability of training data. Green area covers the improvement from our framework.

## F PERFORMANCE ON MULTI-STEP AHEAD PREDICTION TASK

Fig. 10 demonstrates the extended prediction results for KT models for multi-step ahead prediction task with both scenarios: (a) accumulative and (b) non-accumulative prediction. Of note, we excluded IEKT from this task due to its slow inference. As detailed in the implementation section (Sec.D), our KCQRL does not affect the models' original runtimes, so the slow inference is solely attributed to IEKT's original implementation.

Across 14 KT models, our KCQRL improves the prediction performance for both datasets and for both settings. The only exception is sparseKT during the early stages of accumulative prediction on Eedi. As a result, our framework greatly improves long-term predictions of students' learning journeys and their outcomes.

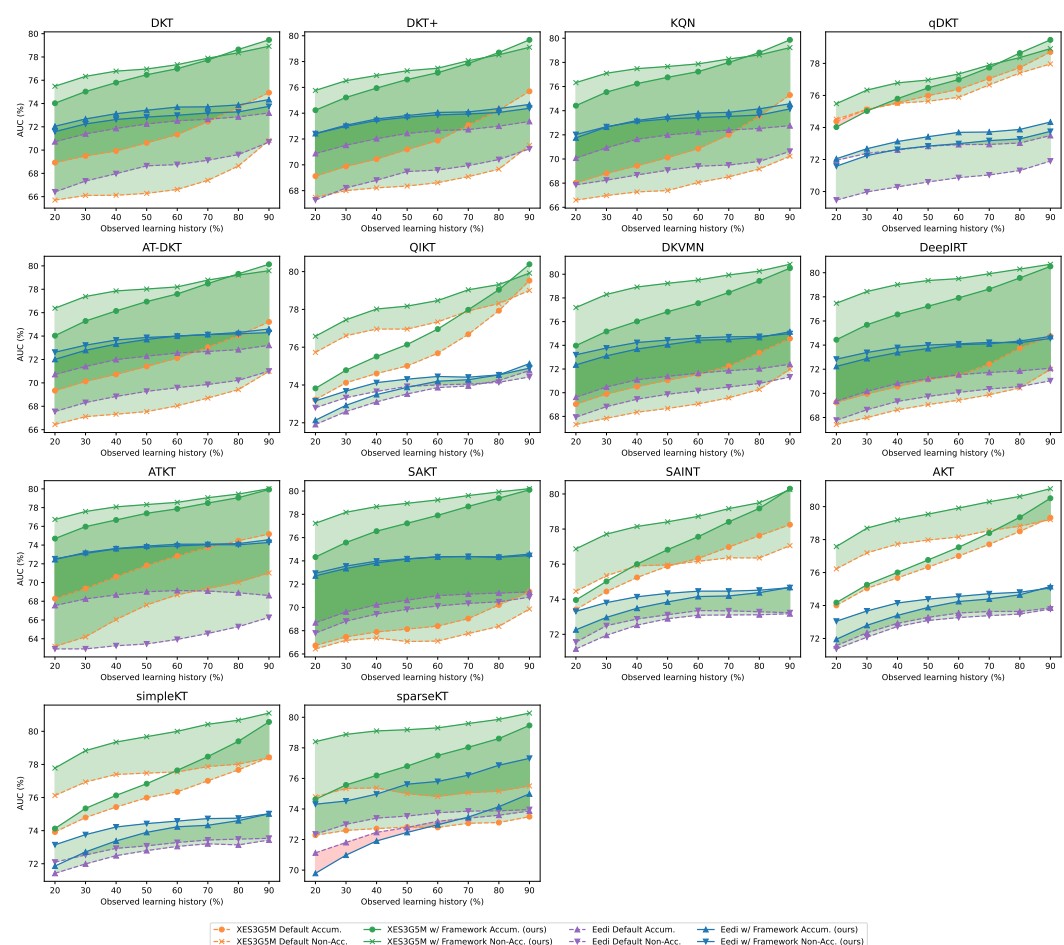

Figure 10: Improvement of our KCQRL across models and datasets in multi-step ahead prediction scenario. Green/red area covers the improvement/decline from our framework.

## G   QUALITY OF AUTOMATED KC ANNOTATION

In this section, we provide the details how we evaluate the quality of our KCQRL's KC annotations as part of our ablation study in Sec. 5.2.

For this, we compared our annotations against two baselines: 1) KC annotations from the original datasets and 2) KC annotations of our KCQRL without leveraging the solution steps.

We picked 1,000 random questions from each dataset, XES3G5M and Eedi. We used Llama-3.1-405B (Dubey et al., 2024) for the evaluations of these questions. Here, we chose a Llama-based model instead of GPT-based models (e. g., GPT-4o) to prevent the potential bias of GPT models towards their own generations, as we already used GPT-4o for our KC annotations earlier.

For a structured comparison, we defined 5 criteria. **Correctness** is to measure which annotation has correct KC(s), relative to the other annotations. **Coverage** is an evaluation criterion for questions with multiple KCs. It measures how well the KCs from an annotation cover the set of KCs that the question is associated with. **Specificity** is a measure to compare which annotation has a more modular set of KCs rather than long and complicated ones. Of note, modular and simpler KCs enable identifying the common skills required for different questions. We additionally defined **ability of integration** to evaluate how well the described KCs are widely applicable to other Math problems, beyond the question being solved. Finally, **overall** is about choosing the best KC annotation by considering all criteria.

As explained in Sec. 5.2, we made a pair-wise comparison between KC annotations: (i) Original vs. KCQRL w/o sol. steps, (ii) Original vs. KCQRL, and (iii) KCQRL w/o sol. steps vs. KCQRL. To eliminate any potential bias from the order of KC annotations in the prompt, each time we randomly assigned KC annotations to groups A and B. Our prompt is given in Appendix G.2.

### G.1   EXAMPLE KC ANNOTATIONS

Here, we follow our example question from Fig. 2 and provide the KC annotations of two baselines and our framework.

**Question:** $65 \cdot 34 + 65 \cdot 45 + 79 \cdot 35 = ?$

**Solution steps:** Below is the extended version of the solution steps, provided by our framework:

- First, factor out the common factor from the first two terms, which is 65. So, the expression becomes $65 \times (34 + 45) + 79 \times 35$.
- Next, simplify the addition inside the parentheses: $34 + 45 = 79$. So, the expression now is $65 \times 79 + 79 \times 35$.
- Notice that 79 is a common factor in both terms, so factor it out: $79 \times (65 + 35)$.
- Simplify the addition inside the parentheses: $65 + 35 = 100$. So, the final result is $79 \times 100 = 7900$.

For the above Math question, the KC annotations are the following:

**Original from the dataset: a)** Extracting common factors of integer multiplication (ordinary type).

**KCQRL w/o solution steps: a)** Understanding of addition. **b)** Ability to perform multiplication.

**Our complete KCQRL framework: a)** Understanding of multiplication. **b)** Understanding of addition. **c)** Factoring out a common factor. **d)** Simplifications of expressions. **e)** Distributive property.

*Takeaway:* The original KC annotation is just one phrase with complex combinations of multiple KCs. It is also missing out some KCs such as addition and simplifications of expressions. On the other hand, KC annotations of KCQRL w/o solution steps are missing the important techniques asked by the problem, such as extracting the common factor. The reason is, this technique is hidden in the solution steps, which need to be provided for a better KC annotation. Overall, our complete framework provides correct set of KCs with better coverage and in a modular way in comparison to two baseline annotations.

## G.2 PROMPT FOR QUALITY COMPARISON OF KC ANNOTATIONS

Below is the prompt for comparing the qualities of different KC annotations via LLMs.

---

**Prompt**

You are provided with a Math question. First, you are asked to solve the given question step by step. Your task is to choose the best knowledge concept (KC) annotation (A or B) for each of the following criteria.

- Correctness: You will choose the KC annotations in terms of the correctness, considering the question and your answer to that question.
- Coverage: KC annotations may contain multiple KCs. If you think the question is originally linked to multiple KCs, choose the KC annotation that covers the most of them. If you think the question is linked to only one KC, choose the KC annotation that covers it. If both covers it, then choose the KC annotation with the last number of KCs.
- Specificity: Choose the least specific KC annotation, ie, consisting of multiple simple elements instead of consisting of a single complex element or a few highly detailed elements. In other words, consider the complexity and specificity of the individual sub-concepts, rather than just the overall number of concepts. A knowledge concept with multiple simple sub-concepts should be ranked as less specific than a knowledge concept with a single, highly detailed sub-concept.
- Ability of integration: Choose the KC annotation that best represents a skill or concept that is widely applicable to other problems or contexts. In other words, select the KC that is more transferable and versatile across various types of math questions, beyond the specific question being solved.
- Overall: Choose the best KC annotation, considering all the metrics above.

The presented KC annotations might have different formats (one with bullet points and the other with new lines etc.). For your evaluation, do not pay attention to the formatting of them, and only focus on their textual content.

These two knowledge concept annotations are given as Group A and B. You will output your selection for each criterion. Please follow the example output (between """s) below as a template when structuring your output.

"""Solution: <Your solution to the Math question>
Correctness: <A or B>
Coverage: <A or B>
Specificity: <A or B>
Ability of integration: <A or B>
Overall: <A or B>"""

Math Question: **<QUESTION CONTENT>**
Knowledge Concept Annotations:
- Group A: **<KC ANNOTATIONS 1>**
- Group B: **<KC ANNOTATIONS 2>**

---

# H    PROMPTS FOR KC ANNOTATION VIA LLMS

Our framework leverages the reasoning abilities of LLMs to annotate the KCs of each question in a grounded manner, which is at the core of our Module 1: KC annotation via LLMs (Sec. 3.1). This is done in three steps. Below, we provide the prompts for each step in order.

## H.1    SOLUTION STEP GENERATION

As the first step, our framework generates the solution steps for the given question. The prompt is given below.

---

**Prompt**

Your task is to generate clear and concise step by step solutions of the provided Math problem. Please consider the below instructions in your generation.

- You will also be provided with the final answer. When generating the step by step solution, you can leverage those information pieces, but you can also use your own judgment.
- It is important that your generated step by step solution should be understandable as stand-alone, meaning that the student should not need to additionally check final answer or explanation provided.
- Please provide your step-by-step solution as each step in a new line. Don't enumerate the steps. Don't put any bullet points. Separate the solution steps only with one newline \n.

Question: **<QUESTION TEXT>**
Final Answer: **<FINAL ANSWER>**
Step by Step Solution:

---

## H.2    KC ANNOTATION

For the given question and its generated solution steps from the earlier part, our KCQRL framework annotates the KCs in a grounded manner. The prompt of this step is given below.

---

**Prompt**

You will be provided with a Math question, its final answer and its step by step solution. Your task is to provide the concise and comprehensive list of knowledge concepts in the Math curriculum required to correctly answer the questions. Please carefully follow the below instructions:

- Provide multiple knowledge concepts only when it is actually needed.
- Some questions may require a figure, which you won't be provided. As the step-by-step solution is already provided, Use your judgment to infer which knowledge concept(s) might be needed.
- For a small set of solutions, their last step(s) might be missing due to limited token size. Use your judgment based on your input and your ability to infer how the solution would conclude.
- Remember that knowledge concepts should be appropriate for Math curriculum between 1st and 8th grade. If the annotated step-by-step solution involves more advanced techniques, use your judgment for more simplified alternatives.

Question: **<QUESTION TEXT>**
Final Answer: **<FINAL ANSWER>**
Step by Step Solution: **<SOLUTION STEPS>**

---

## H.3 Solution Step-KC Mapping

As the final part, our framework maps each solution step to its associated KCs for a given question. This step is particularly needed for the Module 2 of our framework: representation learning of questions (Sec. 3.2). The prompt of this step is given below.

---

**Prompt**

You are expert in Math education. You are given a Math question, its solution steps, and its knowledge concept(s), which you have annotated earlier. Your task is to associate which solution steps require which knowledge concepts. Note that all solution steps and all knowledge concepts must be mapped, while many-to-many mapping is indeed possible.

Each solution step and each knowledge concept is numbered. Your output should enumerate all solution step - knowledge concept pairs as numbers.

Your output should meet all the below criteria:
- Each solution step has to be paired.
- Each knowledge concept has to be paired.
- Map a solution step with a knowledge concept only if they are relevant.
- Your pairs cannot contain artificial solution steps. For instance, If there are 4 solution steps, the pair "5-2" is indeed illegal.
- Your pairs cannot contain artificial knowledge concepts. For instance, If there are 3 knowledge concepts, the pair "3-5" is indeed illegal.

You will output solution step - knowledge concept pairs in a comma separated manner and in a single line. For example, if there are 4 solution steps and 5 knowledge concepts, one potential output could be the following: "1-1, 1-3, 1-5, 2-4, 3-2, 3-5, 4-2, 4-3, 4-5".

Observe that this output also meets all the criteria explained above.

Now, for the given question, solution steps and knowledge concepts, please provide your mapping as the output.

Question: **<QUESTION TEXT>**
Solution steps: **<SOLUTION STEP>**
Knowledge concepts: **<ANNOTATED KCS>**
Solution step - KC mapping:

---

# I SUMMARY STATISTICS OF KC ANNOTATION

For XES3G5M dataset, our KCQRL framework annotated 8378 unique KCs for 7652 questions in total. Fig. 11 shows the distribution of the number KCs annotated per question. Compared to the original dataset with 1.16 KCs per question, our framework identifies 4 or 5 KCs for the majority of questions. It shows that our framework annotates the questions with more modular KCs in comparison to the original dataset.

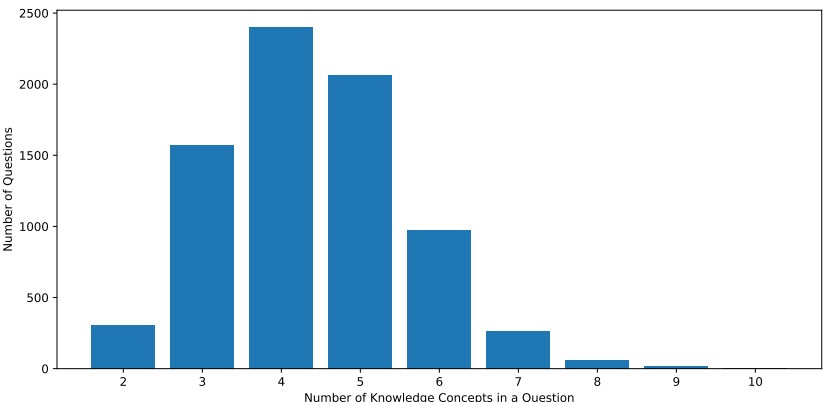

Figure 11: Distribution of questions with different numbers of KCs annotated by our KCQRL framework for XES3G5M dataset.

Our framework identifies 2024 clusters for these 8378 unique KCs annotated. Fig. 12 shows the most frequent clusters across all the questions. To keep the plot informative, we discard the clusters that include basic arithmetic operations as they appear in the majority of the questions. To provide reproducibility and deeper insights into our KC annotations and clusters, we provide the full annotations in our repository.

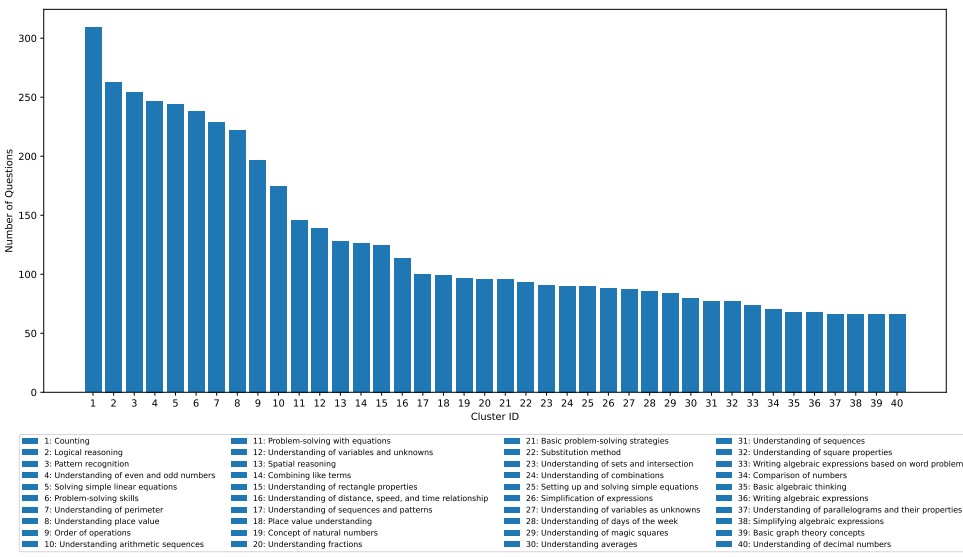

Figure 12: Most frequent KCs annotated across all the questions in XES3G5M dataset. Result is shown after clustering.

## J  QUALITY OF KC ANNOTATION WITH VARYING SIZES OF LLMs

In this section, we inspect the impact of LLM's capacity on the quality of automated KC annotation. For this, we pick different LLMs from the same family of models, namely Llama-3.2-3B, Llama-3.1-8B and Llama-3.1-70B (Dubey et al., 2024). To establish a reference annotation, we further included the KC annotations of the original XES3G5M dataset.

We followed a similar procedure to Appendix G. Specifically, we again picked 1,000 random questions from XES3G5M dataset. As we earlier found that LLMs provide better KC annotations by considering the solution steps (Sec. 5.2 and Appendix G), we provided solution steps to above Llama models for KC annotation. We used the same prompt template as in Appendix H.2. To compare four KC annotations (one original and three Llama models), we leverage GPT-4o instead of any Llama model to prevent the potential bias of Llama models towards their own generations. For comparison, we used the same five evaluation metrics defined in Appendix G), namely correctness, coverage, specificity, ability of integration and overall. We leveraged the same prompt template in Appendix G.2 by only extending it to 4 groups of KC annotations. At each inference, we randomly shuffled the order of KC annotations to avoid a potential bias from the order of KCs presented.

Table 5: **Ablation study showing the relevance of automated KC annotations with varying sizes of LLMs.** We report the quality (in %) for different KC annotations.

|  | Original | Llama-3.2-3B | Llama-3.1-8B | Llama-3.1-70B |
|---|---|---|---|---|
| Correctness | 0.7 | 17.0 | 31.2 | **51.1** |
| Coverage | 1.8 | 19.1 | **47.2** | 31.9 |
| Specificity | 11.5 | 17.4 | 24.6 | **46.5** |
| Ability of integration | 3.3 | 17.1 | 37.3 | **42.3** |
| Overall | 1.3 | 17.1 | 36.1 | **45.5** |

Table 5 compares the quality of KC annotations of varying size of Llama models and the KC annotation of the original dataset. Specifically, it shows % of times where each model is chosen to be the best for each criterion. Overall, larger Llama models are preferred over the smaller ones. The only exception is the coverage where Llama-3.1-8B is found to be the best. After manual inspection, we found that Llama-3.1-8B generates much more KCs than the other models, which leads to increasing coverage but also decreasing correctness and specificity. Further, the overall quality of original KC annotations is only found to be best at only ~1 % of the questions, when compared against three other Llama variants. This further highlights the need for designing better KC annotation mechanisms.

# K    QUALITY OF KC ANNOTATIONS VIA HUMAN EVALUATION

In this section, we conduct a human evaluation to assess the quality of KC annotations. Specifically, we randomly selected 100 questions from the XES3G5M dataset and asked seven domain expert evaluators to compare KC annotations from three sources: **(i)** the original dataset, **(ii)** Llama-3.1-70B[7] (the best-performing model among Llama versions evaluated in Appendix J), and **(iii)** our KCQRL framework leveraging GPT-4o.

To ensure fairness, we randomly shuffled the KC annotations for each question and concealed their sources. For each question, we provided step-by-step solution steps to aid the evaluators' assessments. While we only requested the overall preference of evaluators, we supplied detailed explanations for additional criteria (e. g., correctness, coverage, etc.) to support their decision-making. (Of note, we received the initial feedback that evaluating all five criteria for each question would be too exhaustive, so we proceeded with overall evaluations only.)

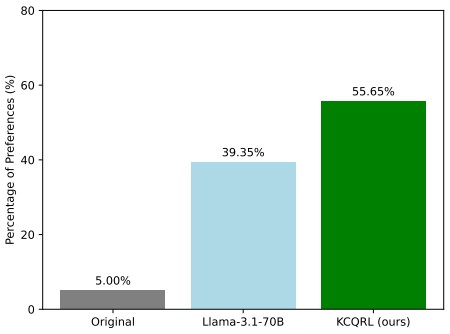

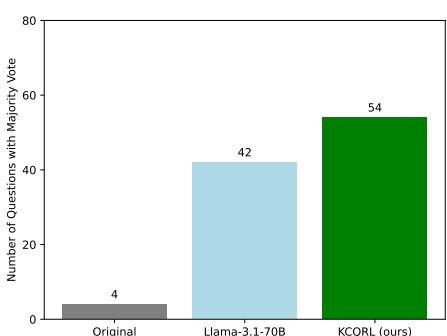

(a) % of Preferences for Different KC Annotations                (b) Majority Vote for Different KC Annotations

Figure 13: Human evaluation on KC annotations

Fig. 13a shows the distribution of preferences for each KC Annotation method, aggregated over all the questions and the evaluators. Only 5 % of the time the evaluators preferred the original KCs over the LLM-generated KC annotations. This percentage is even lower than our automated evaluation of KC annotations in our ablation studies (Sec. 5.2). We also found that the KC annotations of GPT-4o is preferred more than Llama-3.1-70B (55.65 % vs. 39.35 %), which confirms our choice of LLM in KC annotation module (Sec. 3.1).

Fig. 13b shows the number of questions for which each KC annotation method got the majority of the votes. The results are similar to the earlier distribution of preferences. The original KC annotations got the majority of the votes for only 4 questions. In comparison, for more than half of the questions, the annotations of our framework got the majority votes.

We additionally checked if there are any questions where all seven evaluators agreed on the same KC annotation. (Note that, the chance of such agreement at random is only around ∼ 0.14 %.) We found that all evaluators chose GPT-4o for seven questions, and chose Llama-3.1-70B for 3 questions. On the other hand, there is no question for which all seven evaluators preferred original KC annotations. As a result, all our findings from human evaluations show that LLMs are preferred more over the KC annotations from the original dataset. Further, the strong preference over GPT-4o confirms our choice in our KC annotation module.

---

[7]As in Appendix J, the model was provided with solution steps to generate KC annotations, ensuring a fair comparison with our framework.

## L  ABLATION STUDY ON DIFFERENT MODEL EMBEDDINGS

In this section, we design an ablation study to compare the quality of different question embeddings. Specifically, we get the question embeddings from Word2Vec (average over all the words in a sentence), OpenAI's text-embedding-3-small model, and BERT (our encoder LM $E_\psi(\cdot)$ of representation learning module in Sec. 3.2). We follow the same procedure as we compared the question embeddings in Sec. 5.2: we get the embeddings of the question, its solution steps, and KCs concatenated as in the part (d) of Sec. 5.2. We then compare these embeddings based on the performance of downstream KT models. To establish a better reference, we further include the default performance of these KT models (i. e., with random initialized embeddings) and the performance of our KCQRL framework.

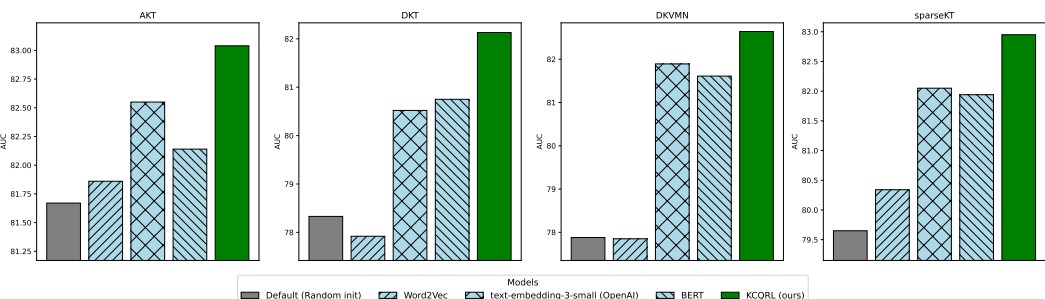

Figure 14: The quality of different embedding methods compared against our KCQRL framework.

Fig. 14 shows the performance of four KT models, AKT, DKT, DKVMN and sparseKT with five different embeddings. We have the following observations: **(1)** The embeddings of Word2Vec does not yield much improvement over the default version, and it might even hurt the performance. We explain this observation by that (i) Word2Vec is not advanced technique to get the semantics of a Math problem and (ii) averaging over the word embeddings can lead question embeddings to concentrate around the same region. **(2)** OpenAI's text-embedding-3-small performs at a similar level to our encoder LM $E_\psi(\cdot)$. The reason is, although performing well on a variety of tasks, this model is not specialized in Math education. **(3)** Our KCQRL framework outperforms other embedding methods, which highlights the need for our representation learning module in Sec. 3.2.

## M  DISCUSSION ON KCQRL USAGE IN REAL-TIME

In this section, we elaborate on how our complete KCQRL framework can be deployed and maintained in real-world scenarios. Below, we discuss various situations that may arise when KCQRL is used for knowledge tracing in online learning platforms.

**What happens when a new student joins the platform?** This scenario has no impact on Module 1 (KC annotation via LLMs in Sec. 3.1) and Module 2 (representation learning of questions in Sec. 3.2) as the database of questions remains unchanged. KT models in our Module 3 (Sec. 3.3) are time-series models that can adapt to new sequences during inference when a new student joins. However, KT models often suffer from cold start issues (Zhang et al., 2014; Fatemi et al., 2023) when students solve only a few exercises. To address this, one can determine the minimum number of exercises required to achieve satisfactory prediction performance, as described in our multi-step ahead prediction analysis in Sec. 5.1. Students can then be guided to follow a curriculum designed by education experts until they complete enough exercises for the KT models to deliver effective predictions.

**What happens when an instructor adds a new question to the platform?** After training the representation learning module (Sec. 3.2), the encoder LM $E_\psi(\cdot)$ learns to associate questions and their solution steps with relevant KCs based on annotations provided by a more capable LLM, such as GPT-4[8]. Consequently, when instructors add a new question to the platform, they can input the question and its solution step into the representation learning module and run the encoder LM $E_\psi(\cdot)$ to generate new embeddings. Details of the inference behavior of $E_\psi(\cdot)$ are provided in Appendix D. To evaluate student knowledge on these new questions, the downstream KT models can then run inference using the newly generated embeddings.

**What happens if the LLM used for KC annotation becomes obsolete?** As mentioned earlier, the LLM for KC annotation is required only during the training of the encoder LM in the representation learning module. Once the encoder LM is trained, our KCQRL framework no longer depends on the LLM from Module 1 during deployment. Therefore, even if the LLM used for KC annotation becomes obsolete, online learning platforms can continue admitting new students and allowing instructors to add new questions. These new questions can still be used to assess student knowledge effectively using our KCQRL framework.

The only exception to this scenario occurs if the platform expands its knowledge tracing to a new subject (e. g., from Math to Chemistry or Physics). In this case, KC annotation for the new question corpus will be necessary to adapt the representation learning module to the new domain via training. If the new subject is unrelated to the existing one (i. e., knowledge in one subject does not inform another), we recommend training and deploying separate branches of the KCQRL pipeline to ensure better performance for the downstream KT models.

**How does KCQRL framework apply to subjects other than Math?** Module 1 (KC annotation via LLMs in Sec. 3.1) in our framework can be extended to other subjects. For subjects where questions do not require multiple solution steps to arrive at the correct answer, the solution step generation part of Module 1 can be skipped, starting directly with the KC annotation process. As demonstrated in our ablation study (Sec. 5.2), our framework still outperforms the original KC annotations even without the inclusion of solution steps. Once KC annotations are obtained, the solution step–KC mapping part of Module 1 can also be omitted, as there are no solution steps to map.

In this scenario, the representation learning module (Sec. 3.2) can be trained solely using $\mathcal{L}_{\text{question}_i}$ (Eq.3). As shown in our ablation study (Sec. 5.2), this approach still improves the performance of downstream KT models. After generating the embeddings, the downstream KT model can then be trained as usual.

Finally, we emphasize that, at the time of writing, KT datasets with available question content are extremely limited; we identified only two such datasets, both in Math. We hope our work highlights the need for KT datasets in other subjects with available question corpora and inspires future research to further integrate question semantics into downstream KT models.

---

[8]This can be viewed as a form of knowledge distillation from a larger LLM to a smaller one.

