# OpenReview forum: "Automated Knowledge Concept Annotation and Question Representation Learning for Knowledge Tracing"
_ICLR.cc/2025/Conference — Submitted to ICLR 2025_

### Official Review · Reviewer_NywH · 2024-11-03

**Soundness:** 3
**Presentation:** 3
**Contribution:** 3
**Rating:** 6
**Confidence:** 2

**Summary:**

This paper proposes a framework named KCQRL, designed to address key limitations in traditional knowledge tracing (KT) models. KT models typically require extensive manual annotation of knowledge concepts (KCs) and often overlook the semantic content of questions. KCQRL introduces automated KC annotation using large language models (LLMs) and employs contrastive learning to create semantically rich question representations. The framework is intended to integrate seamlessly with existing KT models, enhancing performance on prediction tasks across datasets.

**Strengths:**

- The proposed framework’s use of LLMs for KC annotation reduces dependency on manual input, potentially streamlining the annotation process. KCQRL is presented as a flexible solution that can enhance multiple KT models.

- The experimental results across various KT models demonstrate consistent performance gains, supporting the efficacy of integrating semantic embeddings.

- The framework shows potential for low-sample settings, which could benefit platforms with limited data.

**Weaknesses:**

- The quality of KC annotations and question representations is heavily influenced by the LLM used. Reporting the effects of using different LLMs would provide insights into the adaptability of the proposed method. For example, instead of using GPT-4o, employing domain-specific LLMs might enhance the performance?

- The framework's heavy reliance on contrastive learning to handle false negatives might reduce model interpretability, particularly when working with similar KCs.

- Experiments focus solely on two math-focused datasets, making it unclear how well KCQRL would generalize to other subject areas or to question types that are more complex, such as open-ended responses.

**Questions:**

- How does KCQRL handle ambiguous or multi-topic questions that might not map clearly to a single KC?
- What steps, if any, are planned to improve interpretability of the model’s embeddings, especially with similar or overlapping KCs?
- How does KCQRL adapt when LLMs used for KC annotations become outdated or unavailable?

---

> ### Author Response · Authors · 2024-11-20
> **Response to Reviewer NywH (1/n)**
>
> Thank you very much for your thoughtful comments and positive feedback! We improved our paper in several ways by taking your comments and concerns at heart. Specifically, we improved our paper in the following ways:
>
> 1) To address your concern about the quality of KC annotations, we have **provided summary statistics of our KC annotations** (new Appendix I). We further **compared the KC annotations of LLMs with varying sizes** (new Appendix J) and **performed human evaluation to compare the KC annotations between different models and original annotation** (new Appendix K).
>
> 2) For the quality of question representation, **we compared the embeddings of different models** (new Appendix L);
>
> 3) For the usage of our framework on other subjects, and with LLMs used for KC annotation being outdated, we **have added our discussion about the usage of framework** (new Appendix M).
>
> Overall, we would like to thank you for improving the quality of our paper! Please find our detailed response to your comments below:
>
> **W1: The quality of KC annotations and question representations is influenced by the LLM used. Reporting the effects of using different LLMs would provide insights into the adaptability of the proposed method.**
>
> Thank you for your valuable feedback. **We performed additional experiments** to address your concern and improve our paper in the following ways:
>
> 1) We compared LLM quality in KC annotation. For this, we picked different LLMs from the same family of models, namely Llama-3.2-3B, Llama-3.1-8B and Llama-3.1-70B. To establish a reference, we further included the KC annotations of the original dataset. We followed a similar procedure to our Appendix G, where we compared the models based on five criteria, namely, correctness, coverage, specificity, ability of integration and overall.
>
> Overall, *we found that larger Llama models are preferred over the smaller ones*. Further, the overall quality of original KC annotations is found to be best at only ∼1 % of the questions, when compared against three other Llama variants. This further *highlights the need for designing better KC annotation mechanisms*.
>
> 2) We **conducted a human evaluation to assess the quality of KC annotations**. Specifically, we compared the KC annotations of (1) the original dataset, (2) Llama-3.1-70B, which is found to be the best among other Llama models in our earlier experiments, and (3) GPT-4o, the model that we originally used in our KCQRL framework. For this, we randomly selected 100 questions, and asked seven evaluators to compare the KC annotations To ensure fairness, we randomly shuffled the KC annotations for each question and concealed their sources.
>
> We found that *only 5 % of the time the evaluators preferred the original KCs over
> the LLM-generated KC annotations*. In fact, this percentage is even lower than our automated evaluation of KC annotations in Sec. 5.2. We also found that the KC annotations of GPT-4o
> is preferred more than Llama-3.1-70B (55.65 % vs. 39.35 %). This **confirms our choice of LLM in KC annotation module** from Sec. 3.1.
>
> 3) We **compared the embedding quality of different models**. Informed by our ablation study (Fig. 5 and Fig. 6 in Sec. 5.2), we considered the embeddings of “question content + its solution steps + it KCs” as it is found to be performing better. For this textual content, we compared the embeddings of Word2Vec, OpenAI’s text-embedding-3-small model, and BERT. We then compared the performance of downstream KT models. For reference, we also included KT models’ default performance (as a proxy lower-bound) and the performance of our complete framework (as a proxy upper-bound).
>
> We found that: (i) Word2Vec offered little improvement and sometimes hurt performance, likely due to its limitations in capturing Math problem semantics. (ii) *OpenAI’s text-embedding-3-small performs at a similar level to the BERT model*. Our hypothesis is that, although performing well on a variety of tasks, text-embedding-3-small is not specialized in Math education. (iii) Finally, *Our KCQRL framework outperforms other embedding methods*, which **highlights the need for our representation learning module** from Sec. 3.2.
>
> 4) We **provide the summary statistics of our KC annotations in our paper**. Further, we have already provided the full set of KC annotations (and the generated solution steps) in our repository. They can be found in the supplementary material and in our anonymous repository.
>
> In summary, to address your concern, we did the following:
>
> => We have **performed additional experiments to compare the variety of LLM quality in KC annotation** (**new Appendix J**).
>
> => We **conducted human evaluation to assess the quality of KC annotations from LLMs** (**new Appendix K**).
>
> => We have **performed additional experiments to compare the embedding quality of different models** (**new Appendix L**).
>
> => We provided more information about our annotated KCs in our repository and in our **Appendix I**.

---

> ### Author Response · Authors · 2024-11-20
> **Response to Reviewer NywH (2/n)**
>
> **W2: The framework's heavy reliance on contrastive learning to handle false negatives might reduce interpretability.**
>
> Thank you very much for your comment. We argue that our false negative elimination actually **improves** the model interpretability. In Figure 5, we would like to refer to the embedding visualizations of "KCQRL w/o false negative pairs" vs. our full "KCQRL". Without false negatives, the questions sharing similar (but not exactly the same) KCs are pushed apart from each other. This hurts the interpretability of the model embeddings. When false negatives are handled properly, what happens is the following: Two questions with similar KCs are pulled towards their corresponding KCs, and since those similar KCs are also semantically similar to each other (in most cases, if not all), then basically these questions get closer over time. *This could also help retrieving the relevant questions for a given KC, which boosts the interpretability*.
>
> **W3: Experiments focus solely on two math-focused datasets, making it unclear how well KCQRL would generalize to other subjects.**
>
> Thank you for raising your concern. Before we discuss how our KCQRL can generalize to other subjects, we would like to kindly emphasize that KT datasets with available question content are extremely limited. At the time of writing, we could identify only two of such datasets and only in the Math domain. This is why our experiments concentrated on Math-focused datasets.
>
> We hope our work highlights the need for more KT datasets in other subjects with available question corpora. In addition, we hope that our framework inspires future research to further integrate question semantics into downstream KT models in different domains.
>
> Now, we would like elaborate on how our framework can be applied to other subjects:
>
> 1) For the domains of natural sciences such as physics or chemistry, logical and numerical reasoning is still needed to solve the questions. Therefore, our framework can still be used to generate the step-by-step solutions of the given problem and then to annotate the associated KCs as described in our paper. As a final note, one should carefully choose the LLM that has the reasoning capacities for this new domain.
>
> 2) For the domains where step-by-step reasoning is not needed to solve a given problem (eg, asking some facts about History), one can skip the solution step generation part of our Module 1 (Sec 3.1), and can directly start with the KC annotation process. As demonstrated in our ablation study (Sec. 5.2), our framework still outperforms the original KC annotations even without the inclusion of solution steps. Once KC annotations are obtained, the solution step–KC mapping part of Module 1 can also be omitted, as there are no solution steps to map. In that case, our representation learning module (Sec. 3.2) can solely be trained by $L_{question}$ (Eq. 3). As shown in our ablation study (Sec. 5.2) providing the embeddings in this way still improves the performance of downstream KT models.
>
> 3) Alternatively, even for the social sciences where students are asked to provide factually correct answers, LLMs can still be used to extract the explanation/reasoning of the thought process, which could then be used to annotate high quality KCs. We are not aware of any application in education yet; however, in the relation extraction literature [1,2], a similar approach has been used to improve the factual correctness of the LLM answers. We believe a similar approach can be adopted here. In this case, our framework can be used as described in the paper, where the only difference would be the change in the prompting of our solution step generation in Module 1.
>
> => We **added a discussion section in our paper about how our framework can be applied to the other subjects** (**new Appendix M**).
>
> **Q1: How does KCQRL handle ambiguous or multi-topic questions that might not map clearly to a single KC?**
>
> Thank you for your question and for giving us a chance to clarify how our framework has a clear advantage over existing KT approaches, and how it improves the quality of KT datasets. We believe that our framework advances the way of leveraging multiple KCs belonging to the same problem.
>
> - Existing works [1,2,3,4] assume that each question maps to only a **single** KC. This is why, if a given question has multiple KCs, these works serialize the KCs first. For instance, if question q1 has two KCs c1 and c2, and question q2 has two KCs c3 and c4, then these works model the sequence of {q1, q2} as {c1, c2, c3, c4} as if the students solved four questions instead of two. Further, they forced the model to predict the same label for c1 and c2, and c3 and c4 respectively, which cause the information leakage. In contrast, **our KCQRL framework correctly handles the sequence of questions** as {q1, q2}, where the embeddings of these questions are carrying the characteristics of their KCs via our representation learning module (Sec. 3.2).

---

> ### Author Response · Authors · 2024-11-20
> **Response to Reviewer NywH (3/n)**
>
> - Both datasets contain overly complicated descriptions for their KCs, which can easily be divided into multiple simpler KCs. As shown in our Appendix I, our framework identifies 4 or 5 KCs per question whereas the original dataset identifies only 1.16. As we shown in our automated evaluations (Sec. 5.2 and Appendix J) and human evaluations (Appendix K), our KC annotations are preferred much more than the original KC annotations.
>
> Overall, our framework greatly improves the adaptation of questions with multiple KCs both at the data level and at the method level.
>
> **Q2: What steps, if any, are planned to improve interpretability of the model’s embeddings, especially with similar or overlapping KCs?**
>
> As described in our response to W2, we believe that our complete framework actually **improves** the model interpretability. *The advantage of our framework is, if there are two questions with similar KCs, their embeddings will concentrate around the similar region*.
>
> We would like to provide an intuition with a simplified example. Let’s consider question q1 with the KC c1, and consider q2 with the KC c2, where c1 and c2 overlap (i.e., their texts are similar but not the same, and they in fact refer to the same concept). First of all, as our encoder LM is a pre-trained language model, c1 and c2 map to similar embeddings right at the start due to their textual similarity. During the training of our representation learning module, q1 will be pulled by c1, and similarly, q2 will be pulled by c2. **With our false-negative elimination**, the negative pairs (q1, c2) and (q2, c1) will be eliminated. Therefore, q1 will not be pushed by c2 and q2 will not be pushed by c1. As a result, towards the end of the training, the embeddings of q1 and q2 will get close to c1 and c2, just as expected.
>
> As a result of this nice behavior of our representation module, we observe in Figure 5 that the questions with similar KCs get closer in the embedding space.
>
> **Q3: How does KCQRL adapt when LLMs used for KC annotations become outdated or unavailable?**
>
> That's a very good question, and, in fact, our KCQRL has a clear solution in that regard.
>
> In this answer, we are assuming the scenario that an instructor/educator wants to add a new question to the existing dataset, such that the new question is relevant to the curriculum of existing questions (i.e. it belongs to the KC(s) already covered by the dataset.). If your original question involves the violation of our assumption, we would be happy to elaborate on that case too.
>
> In the above case, our trained KCQRL framework does not need any LLM prompting anymore. Therefore, even if the chosen LLM becomes outdated, our KCQRL can function properly. The reason is that our representation learning module (Section 3.2) takes only the question and solution steps at the inference time, where the output embedding is close to the embeddings of its associated KCs, and far from the irrelevant KCs, aligned with the contrastive learning training of our module. In short, if the instructor provides the solution steps along with the new question, one can (a) directly bypass our Module 1, or (b) directly get the embeddings of the question from our Module 2, and 3 and then run the inference of the downstream KT models from Module 3.
>
> We have already provided the script in our repository for properly getting the question embeddings after training the representation learning module.
>
> => For clarity to our readers, we now **added the inference behavior of our representation learning module in our paper** (**new Appendix D**).
>
> => We **added a discussion** about how our framework can keep operating when LLMs used for KC annotations become outdated or unavailable (**new Appendix M**).
>
>
> **References**
>
> [1] Chris Piech, Jonathan Bassen, Jonathan Huang, Surya Ganguli, Mehran Sahami, Leonidas J Guibas, and Jascha Sohl-Dickstein. Deep knowledge tracing. In NeurIPS, 2015.
>
> [2] Aritra Ghosh, Neil Heffernan, and Andrew S Lan. Context-aware attentive knowledge tracing. In KDD, 2020.
>
> [3] Zitao Liu, Qiongqiong Liu, Jiahao Chen, Shuyan Huang, and Weiqi Luo. simpleKT: A simple but tough-to-beat baseline for knowledge tracing. In ICLR, 2023.
>
> [4] Shuyan Huang, Zitao Liu, Xiangyu Zhao, Weiqi Luo, and Jian Weng. Towards robust knowledge tracing models via k-sparse attention. In SIGIR, 2023.

---

### Official Review · Reviewer_h76s · 2024-11-04

**Soundness:** 1
**Presentation:** 3
**Contribution:** 1
**Rating:** 5
**Confidence:** 5

**Summary:**

In this paper, authors bridge the gap by emphasizing the importance of learning question embeddings and propose to automatically annotate the knowledge concepts by using BERT. The idea is somehow recognized, however, the design of both components is not satisfactory. I have commented below with some suggestions for further improvements.

**Strengths:**

1. Traditional methods are more like a sequential prediction task which ignores the semantic information in both Qs and KCs. This paper leverages the power of LLMs to automatically annotate the KCs for each question, and learn a better question embedding guided by the contrastive learning.
2. Figures are well-drawn for readers to follow.
3. Extensive experiments are conducted to support the claim and the importance of question embedding.

**Weaknesses:**

* The authors claim to have a careful negative sampling in this paper to avoid false negatives for contrastive learning. However, the auto-annotation is generated by LLMs which is difficult to control for a consistent and high-quality output all the time. There will definitely be noise introduced or similar concepts annotated in different ways during the annotation which may greatly decrease the performance during semantic learning.
> Authors cluster the similar KCs in the same cluster to reduce the impacts, however, this will still definitely affect the learning performance since they are indeed different KCs in the data level.

> Reversely, I suggest to carefully clean the annotation by asking LLMs to bootstrap a pool or seed bank for standardized KC candidates. Each time, a new KC will be verified with the seeds we already have to avoid chaos, and a checked new one will be added into the bank.

* The design of question embedding is partially recognized for its advantages. However, there is totally nothing to do with the Student history! Seems like authors are just considering them as labels. This is a big problem where in KT tasks, student interactions are very important to guide the personalized prediction based on his/her historical performance. But in this paper, authors completely abandon the semantic or sequence guidance from student histories, which is one of the biggest problems here.

* The notations are confusing, both student and s solution steps are represented as 's'.

* BERT is not an LLM at all. The title is mis-claimed.

**Questions:**

N/A

---

> ### Author Response · Authors · 2024-11-20
> **Response to Reviewer h76s (1/n)**
>
> Thank you for your review. During this rebuttal period, we believe that we greatly improved the quality of our paper in several ways. You can find our detailed response to your comments below:
>
> **W1: The authors claim to have a careful negative sampling in this paper to avoid false negatives for contrastive learning. However, the auto-annotation is generated by LLMs which is difficult to control for a consistent and high-quality output all the time. There will definitely be noise introduced or similar concepts annotated in different ways during the annotation which may greatly decrease the performance during semantic learning.**
>
> We would like to thank you for your comment. We would like to kindly elaborate on how the KC annotation via LLM provides advantages over the existing works.
>
> - **Existing KT methods do not leverage any KC annotation other than the given ids**. As we have shown across 15 KT models and 2 datasets, our KC annotation and the following representation learning module greatly improves the performance of these KT models (Sec. 5).
>
> - **False-negative elimination further boosts the prediction performance.** We first visually showed that our false-negative elimination greatly improves the embeddings of questions, as the questions with similar KCs get closer and create more compact clusters (Fig.5 in Sec. 5.2). Further, we showed that the false-negative elimination improves the performance of downstream KT models as part of our ablation study (Fig. 7 in Sec. 5.2)
>
> - **Our automated evaluations show that LLMs provide better KC annotations that the KCs provided in the original datasets.** We compared the KC annotations across 5 criteria and found that our LLM in Module 1 (GPT-4o) consistently provides KC annotations with better quality than the original ones (Sec. 5.2). During the rebuttal, we have repeated our experiments with various sizes of LLMs, including Llama-3.2-B, Llama-3.1-8B, and Llama-3.1-70B. We found that **even Llama-3.2-3B annotated KCs with the higher quality than the KCs of the original dataset**.
>
> - **Human evaluators showed a strong preference towards LLM annotations.** We randomly selected 100 questions, and asked seven evaluators to compare the KC annotations from the three sources: (1) the original dataset, (2) Llama-3.1-70B, which is found to be the best among other Llama models in our earlier experiments, and (3) GPT-4o, the model that we originally used in our KCQRL framework. To ensure fairness, we randomly shuffled the KC annotations for each question and concealed their sources. **We found that  only 5 % of the time the evaluators preferred the original KCs over the LLM-generated KC annotations.**
>
> Therefore, the quality of the KC annotations via LLMs, and its contribution to the Knowledge Tracing literature cannot be ignored.
>
> Finally, considering that earlier works on Knowledge Tracing did not perform any KC annotation at all, we kindly argue that this point should not be considered a weakness. Instead, we are taking a first step towards the integration of KC annotation and Knowledge Tracing, which, of course, can be improved in future works.

---

> ### Author Response · Authors · 2024-11-20
> **Response to Reviewer h76s (2/n)**
>
> **W2: The design of question embedding is partially recognized for its advantages. However, there is totally nothing to do with the Student history! …. But in this paper, authors completely abandon the semantic or sequence guidance from student histories, which is one of the biggest problems here.**
>
> We would like to kindly clarify a potential misunderstanding here. Our question embeddings are indeed **trained by the student histories**, and we therefore **facilitate personalized prediction**. In short, our framework captures everything that existing KT models do, but in an improved manner via our KC annotation and representation learning modules.
>
> To better explain our contribution, we would like to first refer to the common limitation of existing KT models, which is the "Limitation 2" in our paper. Directly quoting from our paper: "KT models overlook the semantics of both questions and KCs. Instead, they merely treat them as numerical identifiers, whose embeddings are randomly initialized and are learned throughout training. Therefore, existing KT models are expected to “implicitly” learn the association between questions and KCs and their sequential modeling for student histories, simultaneously."
>
> To address this limitation of simultaneous objective, we carefully designed our framework such that we (1) first learn the association between questions and KCs through our representation learning module (Section 3.2) . (2) Then, we "initialize" KT models with those learned embeddings so that KT models learn the sequential modeling for student histories, which is the main purpose of all the proposed time-series models as part of KT. We would like to emphasize the term "initialization" here. As opposed to the earlier works that performed random-initialization of question embeddings, we "initialize" the question embeddings of KT models with our learned embeddings. These embeddings are continued to be trained as part of the training of KT models. Therefore, we are not abandoning  the semantic or sequence guidance from student histories in our framework.
>
> We hope that the above explanation **clarifies why and how our framework is not abandoning student interactions** and we hope it addresses your concern.
>
> **W3: The notations are confusing, both student and s solution steps are represented as 's'.**
>
> Many thanks for your feedback. We now resolved this confusion as ‘s’ is now only used for solution steps.
>
> **W4: BERT is not an LLM at all. The title is mis-claimed**
>
> Our title and section names do not mention BERT being LLM. Therefore, we believe you are referring to our usage of encoder LLM in Section 3.2. If you believe this can cause misunderstandings, we have now changed the naming to "encoder LM" instead. For your information, representation learning module can indeed be implemented by an LLM, such as OpenAI’s text-embedding-3-small model as we show its performance as part of our ablation study in new Appendix L.

---

> > ### Comment · Reviewer_h76s · 2024-11-25
> >
> > Dear authors,
> >
> > Thanks for the detailed rebuttal. I think my major concerns are addressed to some extent. I have raised my score to 5.
> >
> > However, I still think this is just a complete but incremental work. Even though some efforts are new to the community, they are not enough to support this paper to reach bar of ICLR.

---

### Official Review · Reviewer_Mphq · 2024-11-07

**Soundness:** 3
**Presentation:** 3
**Contribution:** 3
**Rating:** 5
**Confidence:** 4

**Summary:**

The authors address two primary shortcomings of existing work on knowledge tracing (KT): (1) mapping questions to knowledge concepts (KCs) being evaluated (I believe from a KC taxonomy defined by subject-matter experts) and (2) a lack of pre-trained embeddings for questions aligned with KCs when applying to KT systems. Methodologically, (1) is addressed by using a SotA LLM to produce a chain-of-thought (CoT)-based answer to a given question and annotate each step with automatically generated KCs and (2) is addressed through a contrastive learning approach to producing questions embeddings that align questions and reasoning steps with KCs produced in (1). Since the learned embeddings are application and not system specific (i.e., it is designed for this task, but not a specific KT model), they can be used as pre-trained embeddings for KT tasks. Accordingly, the authors experiment with adding these to 15 recent KT models with two widely used benchmarks for predicting student response from a sequence of retrospective student interactions, showing consistent improvements over the baseline implementations. Secondary results are provided for the amount of training data (the proposed system needs to see fewer student training examples), quality of the automated KC annotations (generated KCs are better than default annotations as evaluated by a LLM), and the value of the proposed contrastive learning procedure (it is better than embedding generation alternatives). Overall, the authors make a convincing case that the proposed improvements utilizing SotA LLMs advance the SotA for KT problems.

**Strengths:**

This is primarily an applied research submission as it relies on both SotA LLM capabilities (i.e., CoT, knowledge concept generation) within the target domain and existing methods for representation learning with contrastive learning (albeit with domain specific considerations). As previously stated, I think the authors make a convincing case in establishing SotA for the KT setting.

From an originality perspective: (1) other than the task specification and benchmark datasets, there is ostensibly significant methodological innovation within this domain and (2) from a ML/NLP perspective, there is very limited originality; the LLM prompts are relatively straightforward and the contrastive learning procedure follows from a relatively standard 'cookbook'. However, the existing methods are applied correctly, perform well, and introduce new methods to this domain.

From a quality perspective, overall the formulation is well-motivated, conceptually sensible, and performs well. Again, as an applied piece of research, I am convinced that this is advancing the SotA for knowledge tracing. Additionally, the developed methods produce advancements that apply to many existing KT systems (and likely future KT systems) -- and show consistent improvements when applied to existing systems. The 'secondary' experiments are solid from the perspective of supporting the chosen models (e.g., contrastive learning) and answering practical questions with respect to deployment (e.g., number of students in training).

Regarding clarity, the paper is well-motivated, rigorously presented, and easy to understand. The figures are clear overall and any ambiguities can be clarified by the prompts in the appendices. The discussion of experimental results highlights the key findings and makes specific references to table to help the reader focus on the most important quantifications.

Regarding significance, I am not an expert in education, but am convinced that this is a successful application of LLMs to this domain and a notable improvement in performance for the KT task.

**Weaknesses:**

Conversely, below are some of my concerns with this paper:
- From an applied perspective, the experiments are conducted on mathematical reasoning datasets. Since this work is heavily dependent on CoT and KC annotation, it isn't clear that it would work for other domains (e.g., physics, history, etc. -- lines 168-169).
- From a methods perspective, there are limited experiments (e.g., simpler embeddings as a baseline, variety in LLM quality). These aren't mandatory from an applied perspective, but would possible broaden the interest of the paper (which is another potential weakness).
- I am not an education expert, but I am guessing that evaluating KCs with another LLM isn't the standard evaluation procedure. This isn't a big deal since the primary goal of the paper is improving KT results. However, some human-in-the-loop evaluation would help strengthen this specific evaluation.

**Questions:**

The one clarification question I had was regarding if KCs are generally taken from a SME-generated taxonomy and this is relying on (instruction-constrained, but still open-ended) LLM generation. If there is a fixed taxonomy, it would be interesting to see references and/or summary statistics. For the LLM generated cases, it would be interesting to see the statistics of this generated set (e.g., unique labels, perplexity). Clarifying my understanding would be helpful.

The second question is any discussion regarding additional education domains, simpler embeddings, and lower-power LLMs. I don't expect full experiments, but any additional information would be helpful.

---

> ### Author Response · Authors · 2024-11-20
> **Response to Reviewer Mphq (1/n)**
>
> Many thanks for your detailed review and encouraging comments! We have followed your comments very closely and improved our submission in different ways. Specifically,
>
> 1. We have **added summary statistics of our KC Annotation** (new Appendix I),
>
> 2. We have **added new experiments on KC annotation via lower-power LLMs** (new Appendix J).
>
> 3. We have **performed a new human evaluation on KC annotation quality** (new Appendix K).
>
> 4. We have **performed a new ablation study about simple embeddings** (new Appendix L), and
>
> 5. We have **added a new discussion about extending our work additional education domains** and more general discussions about our framework (new Appendix M), all by following your thoughtful comments.
>
> We would like to thank you for your contributions in improving our work!
>
> For your convenience, please find our detailed response to each of your comments below:
>
> **W1: It isn't clear how the proposed framework would work for other domains (e.g., physics, history, etc.)**
>
> Thank you for your comment. We would like to clarify it below for different cases.
>
> 1) For the domains of natural sciences such as physics or chemistry, logical and numerical reasoning is still needed to solve the given questions. Therefore, our framework can still be used to generate the step-by-step solutions of the given problem and then to annotate the associated KCs as described in our paper. As a final note, one should carefully choose the LLM that has the reasoning capacities for this new domain. Unfortunately, *similar datasets for other domains are still missing*, which is an important research gap.
>
> 2) For the domains where step-by-step reasoning is not needed to solve a given problem (eg, asking some facts about History), one can skip the solution step generation part of our Module 1 (Sec 3.1), and can directly start with the KC annotation process. As demonstrated in our
> ablation study (Sec. 5.2), our framework still outperforms the original KC annotations even without the inclusion of solution steps. Once KC annotations are obtained, the solution step — KC mapping part of Module 1 — can also be omitted, as there are no solution steps to map. In that case, our representation learning module (Sec. 3.2) can solely be trained by $L_{question}$ (Eq. 3). As shown in our ablation study (Sec. 5.2) providing the embeddings in this way still improves the performance of downstream KT models.
>
> 3) Alternatively, even for the social sciences where students are asked to provide factually correct answers, LLMs can still be used to extract the explanation/reasoning of the thought process, which could then be used to annotate high quality KCs. We are not aware of any application in education yet; however, in the relation extraction literature [1,2], a similar approach has been used to improve the factual correctness of the LLM answers. We believe a similar approach can be adopted here. In this case, our framework can be used as described in the paper, where the only difference would be the change in the prompting of our solution step generation in Module 1.
>
> Finally, we would like to emphasize that *KT datasets with available question content are extremely limited*. At the time of writing, we could identify only two of such datasets and only in the Math domain. We hope our work highlights the need for KT datasets in other subjects with available question corpora. In addition, we hope that our framework inspires future research
> to further integrate question semantics into downstream KT models in different domains.
>
> => Following your comment, we added a discussion section in our paper about how our framework can be applied to the other domains (**new Appendix M**).
>
> **W2: There are limited experiments (e.g., simpler embeddings as a baseline, variety in LLM quality).**
>
> Thank you very much for your feedback on the scope of our experiments.
>
> - We have added new embeddings to our experiments. Informed by our ablation study (new Fig. 5 and new Fig. 6 in Sec. 5.2). Here, we considered the embeddings of “question content + its solution steps + it KCs” as it is found to be performing better than the other baselines. For this textual content, we compared the embeddings of Word2Vec, OpenAI’s text-embedding-3-small model, and BERT. We then compared the performance of downstream KT models. For reference, we also report the KT models’ default performance (as a proxy lower-bound) and the performance of our complete framework (as a proxy upper-bound).

---

> > ### Author Response · Authors · 2024-11-20
> > **Response to Reviewer Mphq (2/n)**
> >
> > We found that: (i) The embeddings of Word2Vec does not yield much improvement over the default version, and it might even hurt the performance. Our hypothesis is that Word2Vec is not an advanced technique to get the semantics of a Math problem. (ii) OpenAI’s text-embedding-3-small performs at a similar level to the BERT model. Our hypothesis is that, although performing well on a variety of tasks, text-embedding-3-small is not specialized in Math education. (iii) Finally, **our KCQRL framework outperforms other embedding methods**,
> > which *highlights the need for our representation learning module* from Sec. 3.2 .
> >
> > - We have compared the variability of LLM quality in KC annotation. For this, we picked different LLMs from the same family of models, namely Llama-3.2-3B, Llama-3.1-8B and Llama-3.1-70B. To establish a reference annotation, **we further included the KC annotations of the original dataset. We followed a similar procedure in our Appendix G, where we compared the models based on five criteria**, namely, correctness, coverage, specificity,ability of integration and overall.
> >
> > Overall, *we found that larger Llama models are preferred over the smaller ones*.  Further, the overall quality of original KC annotations is only found to be best at only ∼1 % of the questions, when compared against three other Llama variants. This further highlights the need for designing better KC annotation mechanisms.
> >
> > => We have **added new experiments to compare the embedding quality** (**new Appendix L**).
> > => We have now **compared the quality of KC Annotations with varying sizes of LLMs** (**new Appendix J**).
> >
> >
> > **W3: Some human-in-the-loop evaluation would help strengthen this specific evaluation.**
> >
> > Thank you for your valuable feedback. We followed your suggestion, and **conducted a human evaluation to assess the quality of KC annotations**. For this, we randomly selected 100 questions, and asked seven evaluators to compare the KC annotations from the three sources: (1) the original dataset, (2) Llama-3.1-70B, which is found to be the best among other Llama models in our earlier experiments, and (3) GPT-4o, the model that we originally used in our KCQRL framework. To ensure fairness, we randomly shuffled the KC annotations for each question and concealed their sources.
> >
> > We found that  only 5 % of the time the evaluators preferred the original KCs over the LLM-generated KC annotations. In fact, this percentage is even lower than our automated evaluation of KC annotations in Sec. 5.2 . We also found that the KC annotations of GPT-4o
> > is preferred more than Llama-3.1-70B (55.65 % vs. 39.35 %). This **confirms our choice of LLM in KC annotation module in Sec. 3.1**.
> >
> > => We have now **performed a human evaluation to compare the quality of KC annotations**, and provided the full details in our **new Appendix K**.
> >
> > **Q1:  it would be interesting to see the statistics of this generated set of KC annotations.**
> >
> > The KC annotations from our framework are completely LLM generated without following any fixed taxonomy. The reason is that the datasets do not provide the underlying curriculum, and, in addition, different countries follow different standards in their knowledge concept annotations. This is why we did not want to constrain the generations (and also introduce a potential bias) for the KC annotations of LLMs.
> >
> > Following your question, *we provide now the summary statistics of our KC annotations in our paper*. Some statistics are the following (more can be found in the paper): our KCQRL framework annotated 8378 unique KCs (i.e., unique texts) for 7652 questions in total. We have identified 2024 clusters for these 8378 unique KC annotations. Approximately, our cluster are 2.5 times larger than the set of KCs provided by the original dataset (2024 vs. 865). As an important statistic: the original dataset identifies only 1.16 KCs per question, whereas our framework identifies 4 or 5 KCs for the majority of the questions. **This is also another finding confirming our statement: our complete framework provides a correct set of KCs with better coverage and in a modular way**.
> >
> > As a final note, we want to highlight that we have already provided the full set of KC annotations (and the generated solution steps) in our repository.
> >
> > => We have now **added a new section for the summary statistics of KC annotation** (**new Appendix I**).
> >
> > **Q2: Any discussion regarding additional education domains, simpler embeddings, and lower-power LLMs would be helpful.**
> >
> > Thank you very much for all your suggestions! We hope that our discussions regarding additional education domains (**W1 and Appendix M**), our new experiments regarding the simpler embeddings (**W2 and Appendix L**), and our new experiments regarding the lower power LLMs in KC annotation (**W2 and Appendix J**) successfully address your questions.

---

> > > ### Author Response · Authors · 2024-11-20
> > > **Response to Reviewer Mphq (3/n)**
> > >
> > > **References**
> > >
> > > [1] Somin Wadhwa, Silvio Amir, and Byron C Wallace. Revisiting relation extraction in the era of large language models. In ACL, 2023.
> > >
> > > [2] Zhen Wan, Fei Cheng, Zhuoyuan Mao, Qianying Liu, Haiyue Song, Jiwei Li, and Sadao Kurohashi. GPT-RE: In-context learning for relation extraction using large language models. In EMNLP, 2023.

---

### Author Response · Authors · 2024-11-20
**General Response to All Reviewers**

We thank all the reviewers for their time and their helpful comments! We took all the suggestions at heart, and made a substantial revision to our paper. We highlighted the changes in **blue color** (see the rebuttal PDF for download). Our **main improvements** are the following:

1. **We compared the quality of KC annotations with varying sizes of LLMs.** To better observe the impact of the LLM size, we picked the KC annotations from the KT dataset and the annotations from three Llama models with 3B, 8B, and 70B parameters. We run the detailed evaluation for the performance of these models (**new Appendix J**).

2. **We conducted human evaluation to assess the quality of KC annotations.** Specifically, we picked 100 random questions, and asked 7 evaluators to compare the KC annotations of the original dataset, Llama-3.1-70B and GPT-4o in a completely blind manner, with no knowledge of which annotation came from which source. Evaluators chose the LLM annotations ~95% of the time (**new Appendix K**).

3. **We performed experiments with simpler embeddings for the question embeddings**. We further included the performance Word2Vec, OpenAI’s  text-embedding-3-small, and BERT embeddings against our complete KCQRL framework. We found that our representation learning module (Sec. 3.2) clearly improves the performance with respect to these models (**new Appendix L**).

4. **We added a comprehensive discussion section regarding the real-world deployment of our KCQRL framework.** We specifically addressed the questions from our reviewers such as how our framework can be used if the LLM for KC annotation becomes obsolete or how our framework can be extended to domains other than Math. We additionally answered other questions our readers might have (**new Appendix M**)

5. **We further included the summary statistics and key observations about our KC annotations.** Of note, we have already **provided the full set of our generated solution steps and KC annotations in our repository**. In the paper, we aim to provide the key insights to the readers (**new Appendix I**).

With the improvements above, we are confident that we remedied all weaknesses as a result, and are convinced that our paper is a valuable contribution to the literature and a good fit for ICLR 2025. We hope you agree.

---

### Comment · Area_Chair_xeAn · 2024-11-27

Dear reviewers,

Thank you for your efforts reviewing this paper. If you haven't, can you please check the authors' responses and see if your concerns have been addressed? Please acknowledge you have read their responses. Thank you!

---

### Author Response · Authors · 2024-11-29
**Kind Reminder to Reviewers**

Dear Reviewers,

We greatly value the time and effort you’ve dedicated to reviewing our work and engaging in the discussion process. As the discussion period approaches its conclusion, we kindly encourage you to take a moment to review our rebuttal and the improvements made to the manuscript, if you haven’t already.

**We have carefully addressed all the points raised in your initial reviews and made corresponding updates to the paper to reflect these improvements.** Your confirmation on whether our revisions adequately address your concerns would be highly appreciated, and we would welcome any further feedback or suggestions you might have.

Thank you once again for your thoughtful insights and for helping us improve the quality of our work!


Best regards,

The Authors

---

### Meta-Review · Area_Chair_xeAn · 2024-12-21

**Metareview:**

Summary:

This paper presents KCQRL, a framework for automated knowledge concept (KC) annotation and question representation learning to improve the effectiveness of any existing knowledge tracing (KT) model. It proposes an automated KC annotation process using large language models (LLMs), which generates question solutions and then annotates KCs in each solution step of the questions. Then it introduces a contrastive learning approach to generate semantically rich embeddings for questions and solution steps, aligning them with their associated KCs via a tailored false negative elimination approach. These embeddings can be readily integrated into existing KT models, replacing their randomly initialized embeddings. The paper demonstrates the effectiveness of KCQRL across 15 KT algorithms on two large real-world Math learning datasets.

Strengths:

Reviewers generally agree that the paper is well motivated, well presented, and easy to understand. The authors conducted extensive experiments to support the claims on the focused settings.

Weaknesses:

After rebuttal, the following weaknesses still exist:

It is a complete piece of work, but methodological advancements are incremental (Reviewer Mphq, h76s). Focus and experimental scope are a bit narrow, and it is less clear how the proposed framework would work for other domains (Reviewer Mphq, NywH).

**Additional Comments On Reviewer Discussion:**

During the discussion period, the authors have clarified the reviewers’ concerns related to the quality of KC annotations and question representations, the impact of the framework design on model interpretability, human-in-the-loop evaluation, statistics of KC annotations, etc.

---

### Decision · Program_Chairs · 2025-01-22

Reject